# A phosphorylation-regulated NPF transporter determines salt tolerance by mediating chloride uptake in soybean plants

Yunzhen Wu[1,2,6], Jingya Yuan [1,2,6], Like Shen [1,2✉], Qinxue Li[3], Zhuomeng Li[1,2], Hongwei Cao [1,2], Lin Zhu[1,2], Dan Liu[1,2], Yalu Sun[1,2], Qianru Jia [4], Huatao Chen[4], Wubin Wang[2], Jörg Kudla [5], Wenhua Zhang [1,2], Junyi Gai [2✉] & Qun Zhang [1,2✉]

## Abstract

Chloride (Cl⁻) ions cause major damage to crops in saline soils. Understanding the key factors that influence Cl⁻ uptake and translocation will aid the breeding of more salt-tolerant crops. Here, using genome-wide association study and transcriptomic analysis, we identified a NITRATE TRANSPORTER 1 (NRT1)/PEPTIDE TRANSPORTER family (NPF) protein, GmNPF7.5, as the dominant gene locus influencing Cl⁻ homeostasis in soybean (Glycine max). A natural SNP variation resulted in two haplotypes ($GmNPF7.5^{HapA}$ and $GmNPF7.5^{HapB}$), which was associated with Cl⁻ content. $GmNPF7.5^{HapA}$ mediated Cl⁻ or nitrate ($NO_3^-$) uptake in a pH-dependent manner and exhibited higher permeability for Cl⁻ over $NO_3^-$. The suppression of $GmNPF7.5^{HapA}$ expression decreased Cl⁻ accumulation and salt damage in plants, whereas its overexpression showed the opposite effects. The elite haplotype $GmNPF7.5^{HapB}$ diminished Cl⁻ transport activity independently from $NO_3^-$ permeability, thus enhancing soybean salt tolerance. Furthermore, the protein kinase GmPI4Kγ4 could phosphorylate GmNPF7.5, which repressed Cl⁻ uptake without affecting $NO_3^-$ permeability. Our findings define a regulatory mechanism for Cl⁻ control under NaCl stress, providing a strategy for the improvement of salt tolerance in soybean plants.

**Keywords** Chloride Homeostasis; NPF Transporters; Phosphorylation; Soybean; Salt Tolerance
**Subject Category** Plant Biology

## Introduction

Soil salinity significantly restricts global crop productivity and represents a critical threat to sustainable agriculture development and food security (Munns and Tester, 2008; Zhu, 2016). Soybean (*Glycine max*) is a major oil crop worldwide (Zhang et al, 2019); however, salinity stress threatens to dramatically reduce soybean production. Therefore, exploring the mechanism by which soybean responds to salt stress is essential for developing salt-tolerant soybean varieties and ensuring the sustainability of the global food supply.

Salt stress damages crops mainly through osmotic stress and ion toxicity, as a result of high concentrations of sodium (Na⁺) and chloride (Cl⁻) (Yang and Guo, 2018). Na⁺ toxicity is most commonly associated with negative effects induced by salt stress, such that the terms "Na⁺ stress" and "salt stress" are often used interchangeably (Bazihizina et al, 2019; Lacan and Durand, 1995; Munns and Tester, 2008). However, increasing evidence indicates that excessive Cl⁻ accumulation in plants also results in osmotic stress, toxic reactive oxygen species bursts, and inhibition of photosynthesis (Christoph, 2018; Ren et al, 2021; Tavakkoli et al, 2011). Moreover, Cl⁻ ions can restrict the uptake and transport of $NO_3^-$ due to competitive inhibition, as they can share the same anion transporters (Ashraf et al, 2018; Glass and Siddiqi, 1985; Li et al, 2017). Visual Cl⁻ toxicity symptoms include chlorosis and marginal necrosis in leaves, as well as declines in fertilization and yield (Christoph, 2018; Teakle and Tyerman, 2010). Notably, for some plants, such as rough lemon (*Citrus × jambhiri*), common grape vine (*Vitis vinifera*), soybean (*Glycine max*), and narrowleaf trefoil (*Lotus tenuis*), Cl⁻ accumulation is more significantly correlated with symptoms of NaCl stress than of Na⁺ stress (Li et al, 2017; Wu and Li, 2019; Yin et al, 2023). However, numerous studies have focused on the effects of Na⁺/K⁺ transporters on ionic homeostasis and salt tolerance in various species (Bazihizina, 2019; Lacan and Durand, 1995; Zhu, 2016), whereas the regulatory mechanism of Cl⁻ homeostasis in response to salt stress has seldom been investigated.

¹College of Life Sciences, Nanjing Agricultural University, 210095 Nanjing, China. ²State Key Laboratory of Crop Genetics & Germplasm Enhancement and Utilization, National Center for Soybean Improvement, Key Laboratory for Biology and Genetic Improvement of Soybean (General, Ministry of Agriculture), Jiangsu Collaborative Innovation Center for Modern Crop Production, Nanjing Agricultural University, 210095 Nanjing, China. ³Provincial International Science and Technology Cooperation Base on Engineering Biology, International Campus of Zhejiang University, 314400 Hangzhou, China. ⁴Institute of Industrial Crops, Jiangsu Academy of Agricultural Sciences, 210014 Nanjing, China. ⁵Institut für Biologie und Biotechnologie der Pflanzen (IBBP), Universität Münster, Münster, Germany. ⁶These authors contributed equally: Yunzhen Wu, Jingya Yuan. ✉E-mail: likeshen@njau.edu.cn; sri@njau.edu.cn; zhangqun@njau.edu.cn

Several types of $Cl^-$ channel or transporter have been identified in plants, such as the chloride channel (CLC) (Rajappa et al, 2024), aluminum-activated malate transporter (ALMT) (Baetz et al, 2016; De Angeli et al, 2013), slow-type anion channel (SLAC)/SLAC-associated homolog (SLAH) family, cation–$Cl^-$ cotransporters (CCCs) (Colmenero-Flores et al, 2007), NITRATE TRANSPORTER 1 (NRT1)/PEPTIDE TRANSPORTER family (NPF) (Li et al, 2016; Wen et al, 2017; Xiao et al, 2021), and the multidrug and toxic compound extrusion (MATE) family (Yin et al, 2023; Zhang et al, 2014). Among these $Cl^-$ transporters, NPF transporters are receiving increasing attention, possibly due to their dominant contribution to nutrient nitrate ($NO_3^-$) transport, which is antagonistic to $Cl^-$ transport under certain conditions. In *Arabidopsis*, AtNPF2.4 mediates $Cl^-$ loading into root xylem, whereas AtNPF2.5 in root cortex excludes $Cl^-$ out of roots (Li et al, 2016). AtNPF6.3 and its maize homolog ZmNPF6.6 can transport $Cl^-$ in the absence of $NO_3^-$, whereas ZmNPF6.4 mediates the uptake of both $Cl^-$ and $NO_3^-$ but shows a strong preference for $Cl^-$ (Wen et al, 2017). In *Medicago* species, MtNPF6.5 is also more selective for $Cl^-$ over $NO_3^-$ and determines plant $Cl^-$ accumulation (Xiao et al, 2021). Besides $NO_3^-$ and $Cl^-$, NPF proteins have been shown to transport a broad variety of substrates, including glycerate (Lin and Tsay, 2023), abscisic acid (Zhang et al, 2022), auxin (Watanabe et al, 2020), gibberellins (Binenbaum et al, 2023), α-tomatine (Kazachkova et al, 2021), and others (Chao et al, 2021; Michniewicz et al, 2019; Payne et al, 2017). Although soybean contains 115 NPFs, which are divided into eight subfamilies (Léran et al, 2014), the roles of most soybean NPF proteins remain unclear, particularly in the regulation of $Cl^-$ homeostasis under salt stress.

To identify the dominant locus affecting $Cl^-$ homeostasis in soybean, we conducted a genome-wide association study (GWAS) and transcriptomic analysis, and identified an NPF protein (GmNPF7.5) whose single-nucleotide polymorphisms (SNPs) variation determined $Cl^-$ accumulation and plant salt tolerance. GmNPF7.5 was found to mediate $Cl^-$ and $NO_3^-$ uptake in *Xenopus* oocytes but preferentially selected $Cl^-$. The facilitation of $Cl^-$ accumulation by GmNPF7.5 decreased salt tolerance in soybean plants. We also identified GmPI4Kγ4, a protein kinase that phosphorylated GmNPF7.5, which functions as a molecular switch in the inhibition of GmNPF7.5-mediated $Cl^-$ uptake.

# Results

## Identification of a *GmNPF* gene associated with salt tolerance in soybean plants

To identify the key genes related to $Cl^-$ accumulation in soybean plants, we measured shoot $Cl^-$ content in a soybean population composed of 198 accessions under salt (NaCl) stress conditions. Then we conducted a GWAS to identify SNPs associated with shoot $Cl^-$ content (Fig. 1A,B). We also performed whole-transcriptome RNA sequencing (RNA-seq) to identify genes that responded to NaCl stress. In total, 10,029 differentially expressed genes (DEGs) were obtained after salt treatment, among which 4816 genes were upregulated. Notably, 47 candidate genes identified in the GWAS analysis were also found to be upregulated by NaCl stress according to RNA-seq (Fig. 1C). Quantitative reverse-transcription polymerase chain reaction (qRT-PCR) was conducted to verify these

changes in DEG expression, using a threshold of $\log_2(|$fold change$|) \geq 3$; GmNPF7.5 (Glyma.18G260000) showed the highest upregulation in response to salt stress (Figs. 1D and EV1A). This was classified into two haplotypes according to three SNPs in the coding sequence (Fig. 1F,G). The SNP variation in G1735A caused an amino acid substitution conferring $Val^{579}$ to $Ile^{579}$. The HapA (SNP1735-G) and HapB (SNP1735-A) SNPs were associated with higher and lower $Cl^-$ levels, respectively, evaluated by the ratio of $Cl^-$ content under NaCl stress to that under control conditions (Fig. 1H).

The expression patterns of GmNPF7.5 were investigated through β-glucuronidase (GUS) staining of transgenic hairy roots harboring *ProGmNPF7.5:GUS*. Under non-stress conditions, GmNPF7.5 was mainly expressed in root stele tissues (Fig. 1E). Following NaCl treatment, its expression was significantly induced in the root cortex and epidermis (Fig. 1E). To investigate whether NaCl-induced expression was associated with $Cl^-$ or $Na^+$, we treated soybean plants with different salts. Its expression was significantly enhanced upon treatment with 100 mM NaCl or KCl but not with 50 mM $Na_2SO_4$ (Fig. EV1B). These results indicate that GmNPF7.5 is induced by excessive $Cl^-$ and may be the key gene associated with both $Cl^-$ levels and plant salt tolerance.

## GmNPF7.5 negatively regulates salt tolerance in soybean

To further investigate the role of GmNPF7.5 in salt tolerance, we generated chimeric plants carrying transgenic hairy roots characterized by GmNPF7.5 overexpression (GmNPF7.5-OE) or knockdown (GmNPF7.5-RNAi) in "Williams 82" (a cultivar with to HapA) (Appendix Fig. S1A). Compared to the empty vector (EV) control, GmNPF7.5 knockdown decreased stress damage in plants, whereas its overexpression induced the opposite effects under NaCl or KCl treatment (Fig. 2A,B). These phenotypes were associated with $Cl^-$ concentrations in the roots and shoots (Fig. 2C,D). GmNPF7.5-OE plants accumulated more $Cl^-$, whereas GmNPF7.5-RNAi plants accumulated less $Cl^-$ compared to EV control plants. There were no significant differences among the genotypes under $Na_2SO_4$ treatment (Fig. 2A–D).

As high $Cl^-$ concentrations inhibit $NO_3^-$ absorption and nitrogen metabolism under salt stress (Bazihizina et al, 2019), we measured $NO_3^-$ levels in roots and shoots (Appendix Fig. S1B,C). All plants had similar $NO_3^-$ concentration in roots and shoots under control conditions, whereas $NO_3^-$ concentrations significantly decreased following NaCl or KCl treatment. Compared to the EV control, GmNPF7.5-OE plants had lower $NO_3^-$ content, and GmNPF7.5-RNAi plants showed higher $NO_3^-$ accumulation (Appendix Fig. S1B,C). No significant differences in $NO_3^-$ concentrations were found among genotypes upon $Na_2SO_4$ treatment. After NaCl stress, the effects of GmNPF7.5 on $Cl^-$ and $NO_3^-$ concentrations in xylem sap were consistent with those on shoot or root $Cl^-$ and $NO_3^-$ concentrations (Fig. EV2), suggesting that GmNPF7.5 may mediate $Cl^-$ uptake, resulting in enhanced $Cl^-$ accumulation, but decreased $NO_3^-$ accumulation, under salt stress.

To investigate the effect of SNP variation in G1735A on soybean salt tolerance, we generated chimeric plants with transgenic hairy roots overexpressing two haplotypes of GmNPF7.5 (*GmNPF7.5^{HapA}*-OE and *GmNPF7.5^{HapB}*-OE) (Appendix Fig. S1D). *GmNPF7.5^{HapB}*-OE plants were less sensitive to salt stress than *GmNPF7.5^{HapA}*-OE plants (Fig. 2E,F), which was consistent with $Cl^-$ accumulation

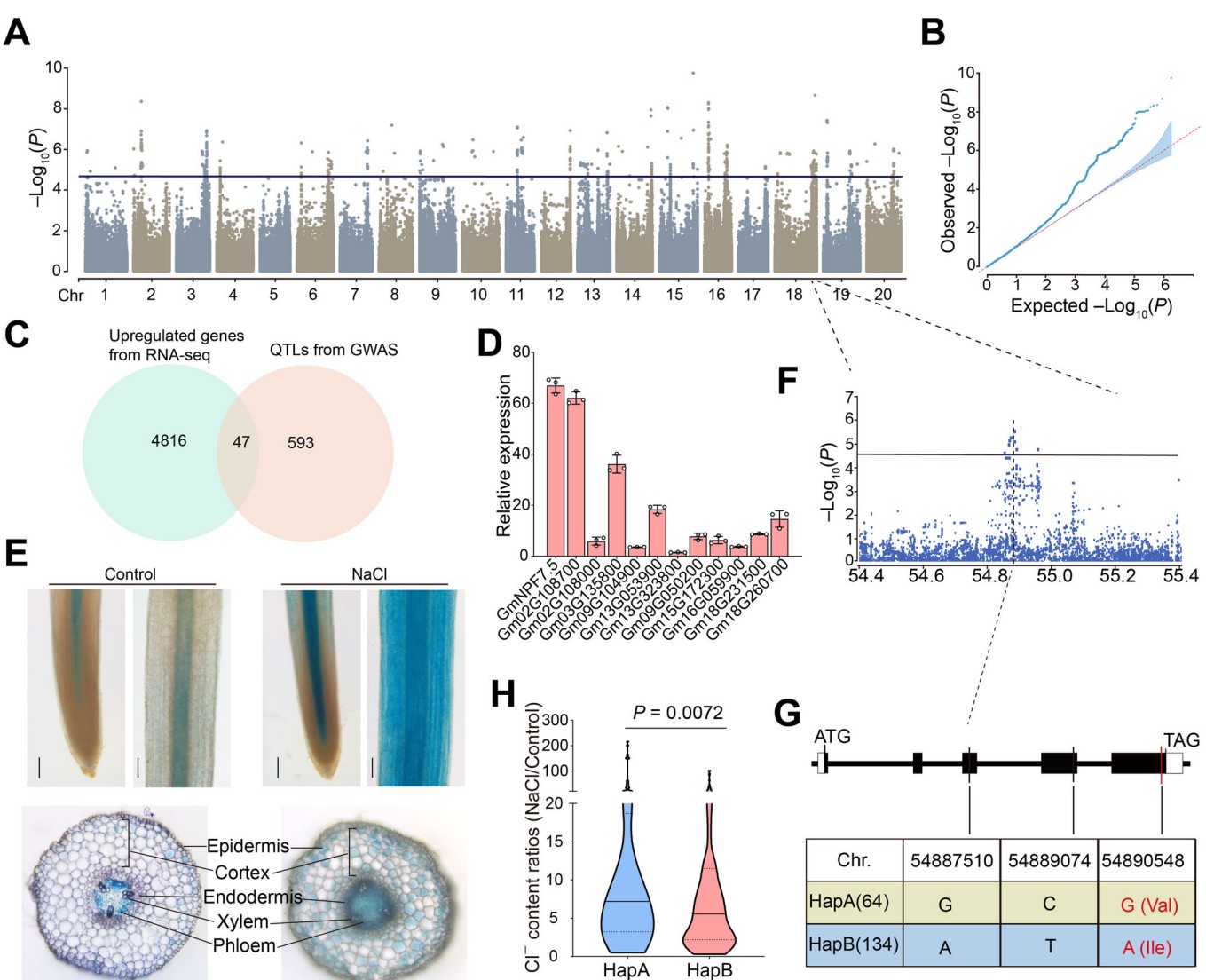

**Figure 1. Identification and haplotype analysis of *GmNPF7.5* associated with excessive chloride (Cl⁻) accumulation in soybean plants under salt stress.**

(A) Results of a genome-wide analysis study (GWAS) of shoot Cl⁻ content under salt stress (150 mM NaCl). Horizontal solid lines indicate the GWAS significance threshold ($-\log_{10}(P) > 4.6$). (B) Quantile–quantile plot of observed vs. expected $P$ values. Both axes were $-\log_{10}$-transformed. The GWAS mixed linear model was run using the FaST-LMM program (Lippert et al, 2011), with the threshold for significant association set to $1/n$, where $n$ is the effective number of independent SNPs ($P < 2.43 \times 10^{-5}$ or $-\log_{10}(P) > 4.6$). (C) Venn diagrams of differentially expressed genes (DEGs) under salt stress and quantitative trait loci (QTLs) related to Cl⁻ content. (D) Quantitative reverse-transcription polymerase chain reaction (qRT-PCR) analysis of salt stress DEGs and QTLs related to Cl⁻ content. *GmELF* was used as an internal control. Data are means ± standard error (SEM). $n = 3$ (3 biological replicates). (E) β-glucuronidase (GUS) staining of *proGmNPF7.5:GUS* transgenic hairy roots. For NaCl treatment, 2-week-old seedlings were treated with 150 mM NaCl for 6 h. In each condition, the left panels showed the root tip zones containing root cap, cell division zone, elongation zone and maturation zone. The right panels showed the middle part of roots, which also belonged to maturation zone. The cross-sections were made using the roots of maturation zone. Scale bars, 200 μm. (F) Manhattan plot showing the Cl⁻ content QTL region (54.4–55.4 Mb) on chromosome 18. The GWAS mixed linear model was run using the FaST-LMM program (Lippert et al, 2011), with the threshold for significant association set to $1/n$, where $n$ is the effective number of independent SNPs ($P < 2.43 \times 10^{-5}$ or $-\log_{10}(P) > 4.6$). (G) Gene structure of *GmNPF7.5* and two haplotypes (HapA and HapB) in 198 soybean accessions. Gray and black boxes indicate untranslated regions (UTRs) and exons, respectively. The single-nucleotide polymorphism (SNP) at 54890548G to A (1735 in the *GmNPF7.5* coding sequence) confers Val[579] to Ile[579]. (H) Comparison of relative Cl⁻ content between the two haplotypes. Significance was evaluated using a two-sided Student's $t$ test ($n = 64$ accessions (each accession contained one independent biological replicate) for genotype HapA; $n = 134$ accessions (each accession contained one independent biological replicate) for genotype HapB). Source data are available online for this figure.

patterns in roots and shoots (Fig. 2G,H). By contrast, NO₃⁻ concentrations were higher in *GmNPF7.5^HapB*-OE plants than in *GmNPF7.5^HapA*-OE plants (Appendix Fig. S1E,F). Together, these results indicate that SNP variation in G1735A determines soybean salt tolerance by regulating the accumulation of Cl⁻ and NO₃⁻ in plants.

## GmNPF7.5 can transport Cl⁻ and NO₃⁻

To determine the transport properties of GmNPF7.5, we expressed GmNPF7.5 in *Xenopus laevis* oocytes for two-electrode voltage clamp recording. GmNPF7.5 fused with green fluorescent protein (GFP) localized at the plasma membrane of oocytes, and similar

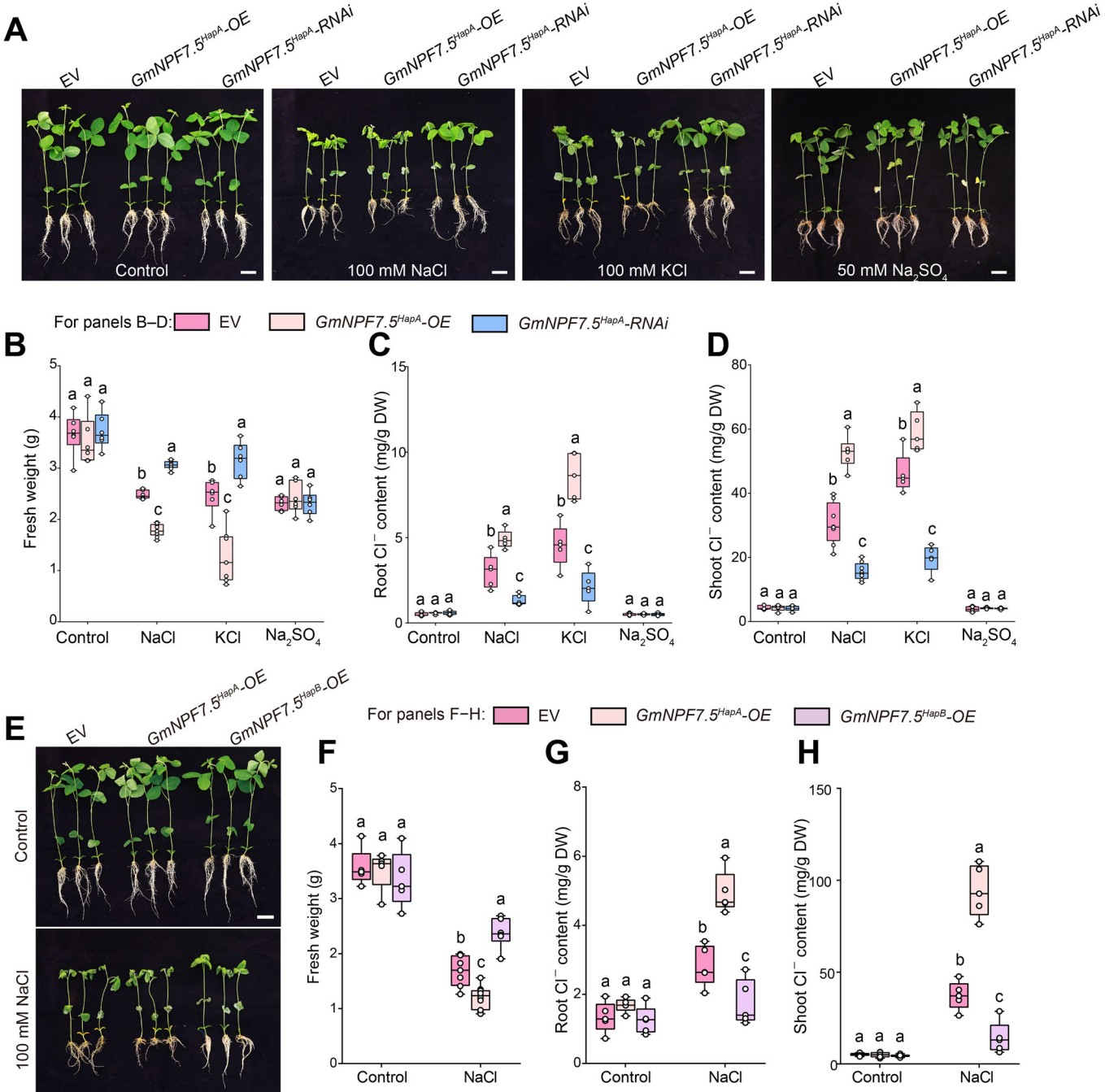

localization was observed in *Nicotiana benthamiana* leaves (Fig. EV3A,B). The GmNPF7.5-expressed oocytes showed significant voltage-dependent currents when exposed to 10 mM $Cl^-$ at pH 5.5 (Fig. 3A), at levels similar to currents observed in oocytes expressing $Cl^-$ transporter ZmNPF6.6 (Wen et al, 2017). By contrast, $Cl^-$-elicited currents were dramatically inhibited at pH 7.5 (Fig. 3B). We verified these results by determining the $Cl^-$ concentrations in *GmNPF7.5*-expressed oocytes (Fig. 3C); the results confirmed that GmNPF7.5 mediated $Cl^-$ uptake, coupled with the influx of $H^+$. Furthermore, $Cl^-$-elicited currents of GmNPF7.5 increased as external $Cl^-$ concentrations increased at pH 5.5 (Fig. 3D–F).

GmNPF7.5 also mediated the symport of $NO_3^-$ and $H^+$ in oocytes exposed to 10 mM $NO_3^-$ (Fig. 3G,H). The NPF family is divided into high- and low-affinity $NO_3^-$ transport systems (Wen et al, 2017). AtNPF6.3 is a dual-affinity $NO_3^-$ transporter that functions under both high (10 mM) and low (0.25 mM) $NO_3^-$ conditions (Liu et al, 1999). When supplied with 0.25 mM $NO_3^-$, GmNPF7.5 did not mediate $NO_3^-$ uptake, whereas AtNPF6.3, the positive control, accumulated more $NO_3^-$ than water-injected oocytes. When supplied with 10 mM $NO_3^-$, both GmNPF7.5 and AtNPF6.3 showed $NO_3^-$ uptake activity (Fig. EV3C,D). In addition, GmNPF7.5[HapB] showed significantly lower $Cl^-$ transport activity compared to GmNPF7.5[HapA] at pH 5.5 (Fig. 3I), whereas there were

◀ **Figure 2. GmNPF7.5, specifically involved in the Cl⁻ stress response, negatively regulated soybean salt tolerance.**

(A) Phenotypic comparison of soybean plants with transgenic hairy roots harboring *GmNPF7.5^HapA* overexpression (*GmNPF7.5^HapA*-OE) or knockdown (*GmNPF7.5^HapA*-RNAi), or empty vector (EV). Scale bars, 5 cm. (B–D) Fresh weight (B), root Cl⁻ content (C), and shoot Cl⁻ content (D) of soybean plants under the indicated conditions. Data in (B–D) are means ± standard error (SEM) ($n = 5$–10 independent biological replicates). Significance was determined using one-way analysis of variance (ANOVA), followed by Tukey's test. Different letters indicate significant differences ($P < 0.05$). (B) Control: $P$ values = 0.7321 (EV vs *GmNPF7.5^HapA*-OE), 0.9357 (EV vs *GmNPF7.5^HapA*-RNAi), 0.5163 (*GmNPF7.5^HapA*-OE vs *GmNPF7.5^HapA*-RNAi). (The following is the same order). NaCl: $P$ values = 0.0003, 0.0026, <0.0001. KCl: $P$ values < 0.0001, 0.001, <0.0001. Na₂SO₄: $P$ values = 0.8128, >0.9999, 0.8095. (C) Control: $P$ values = 0.9803, 0.9812, >0.9999. NaCl: $P$ values = 0.0002, 0.0004, <0.0001. KCl: $P$ values < 0.0001, <0.0001, <0.0001. Na₂SO₄: $P$ values = 0.9987, >0.9999, 0.999. (D) Control: $P$ values = 0.998, 0.9956, 0.9995. NaCl: $P$ values < 0.0001, <0.0001, <0.0001. KCl: $P$ values < 0.0001, <0.0001, <0.0001. Na₂SO₄: $P$ values = 0.9956, 0.997, 0.9999. (E) Phenotypic comparison of soybean plants with transgenic hairy roots harboring a haplotype of *GmNPF7.5*-OE or empty vector (EV). Scale bars, 5 cm. (F–H) Fresh weight (F), root Cl⁻ content (G), and shoot Cl⁻ content (H) of soybean plants under control and NaCl treatments. Data in (F–H) are means ± standard error (SEM) ($n = 5$–10 independent biological replicates). Significance was determined using one-way analysis of variance (ANOVA), followed by Tukey's test. Different letters indicate significant differences ($P < 0.05$). (F), Control: $P$ values = 0.9713 (EV vs *GmNPF7.5^HapA*-OE), 0.5289 (EV vs *GmNPF7.5^HapB*-OE), 0.6707 (*GmNPF7.5^HapA*-OE vs *GmNPF7.5^HapB*-OE). (The following is the same order). NaCl: $P$ values = 0.0134, 0.0011, <0.0001. (G) Control: $P$ values = 0.5407, 0.9558, 0.3782. NaCl: $P$ values < 0.0001, 0.0081, <0.0001. (H) Control: $P$ values = 0.9993, 0.9885, 0.9933. NaCl: $P$ values < 0.0001, 0.0002, <0.0001. Data in (B–D, F–H) are plotted with box–whisker plots: the whiskers represent maximum and minimum values, and boxes represent the upper quartile, median, and lower quartile, dots represent data points. Source data are available online for this figure.

no significant differences in NO₃⁻ transport activity (Fig. 3J), suggesting that SNP variation (G1735A) diminished Cl⁻ transport activity without affecting NO₃⁻ permeability. This mechanism may underlie the lower accumulation of Cl⁻ and enhanced salt tolerance in GmNPF7.5^HapB-OE plants compared to GmNPF7.5^HapA-OE plants (Fig. 2E–H). Furthermore, GmNPF7.5-mediated NO₃⁻ transport was significantly reduced in the presence of Cl⁻, whereas NO₃⁻ did not affect the Cl⁻ transport activity of GmNPF7.5 (Fig. 3K,L). These findings suggest that GmNPF7.5 may exhibit greater selectivity for Cl⁻ over NO₃⁻.

## GmPI4Kγ4 interacts with GmNPF7.5 and enhances soybean salt tolerance by inhibiting Cl⁻ transport

The yeast two-hybrid system was used to screen for proteins interacting with the central linker (CL) domain of GmNPF7.5 (GmNPF7.5CL) (Fig. 4A; Appendix Fig. S2A). Among the candidates, the phosphatidylinositol-4-kinase GmPI4Kγ4 (Glyma.16G02400) caught our attention, because GmPI4Kγ4–GFP was localized at the plasma membrane, cytoplasm, and nucleus in *N. benthamiana* leaves (Appendix Fig. S2B), and salt stress significantly induced the expression of *GmPI4Kγ4* (Appendix Fig. S2C).

An in vitro pull-down assay showed that GmPI4Kγ4 interacted with GmNPF7.5CL tagged with glutathione-S-transferase (GST) (Fig. 4B). Bimolecular fluorescence complementation (BiFC) assays indicated that GmNPF7.5 interacted with GmPI4Kγ4 at the plasma membrane in *N. benthamiana* leaves (Appendix Fig. S3A). A coimmunoprecipitation (Co-IP) assay also showed that GmNPF7.5CL was coimmunoprecipitated with GmPI4Kγ4 (Fig. 4C). Furthermore, we conducted Förster resonance energy transfer by fluorescence lifetime imaging (FRET–FLIM) to detect the interaction between GmNPF7.5–GFP and GmPI4Kγ4-mCherry in *N. benthamiana* leaves (Fig. 4D). The fluorescence lifetime of GmNPF7.5–GFP/GmPI4Kγ4-mCherry was significantly lower than that of the GmNPF7.5–GFP/AtCBL1n–mCherry combination (Fig. 4E), and salt stress enhanced their interaction (Fig. 4D,E). Together, these results indicate that GmNPF7.5 interacts with GmPI4Kγ4 in plants.

Next, we overexpressed *GmPI4Kγ4* in soybean plants with or without GmNPF7.5 overexpression (Appendix Fig. S3B). *GmPI4Kγ4* overexpression in wild-type (WT) plants improved soybean salt tolerance, and simultaneous overexpression of both

GmNPF7.5 and *GmPI4Kγ4* significantly alleviated the salt-sensitive phenotypes induced through *GmNPF7.5* overexpression (Fig. 4F–H). Furthermore, *GmPI4Kγ4* overexpression significantly decreased Cl⁻ accumulation and enhanced NO₃⁻ levels in roots (Fig. 4I–L), indicating that GmPI4Kγ4 positively regulates soybean salt tolerance by inhibiting GmNPF7.5-mediated Cl⁻ transport.

## GmPI4Kγ4 phosphorylates GmNPF7.5

Given that GmPI4Kγ4 interacts with GmNPF7.5, we examined whether GmPI4Kγ4 could directly phosphorylate GmNPF7.5 through in vitro kinase assays on recombinant GST-tagged GmPI4Kγ4 using the GmNPF7.5CL fragment as a substrate. GmPI4Kγ4 indeed phosphorylated GmNPF7.5CL (Fig. 5A). Similar phosphorylation by GmPI4Kγ4 also occurred for GmNPF7.10 but not for GmNPF7.11, although both are close homologs of GmNPF7.5 (Fig. 5B; Appendix Fig. S4A). According to sequence alignment differences among the three NPFs, we speculated that four residues (Ser295, Thr296, Ser336, and Thr337) could be the relevant phosphorylation sites (Appendix Fig. S4B). All four phosphorylated sites were located in the intracellular (central) linker domain separating two transmembrane domain bundles (Fig. 5C). Next, we generated a mutant construct with these four residues substituted by alanine (GmNPF7.5^S295A/T296A/S336A/T337A, abbreviated GmNPF7.5^4A); we detected no phosphorylation of GmNPF7.5^4A in the presence of GmPI4Kγ4 (Fig. 5D).

The co-expression of GmPI4Kγ4 and GmNPF7.5 in oocytes significantly inhibited the Cl⁻ transport activity of GmNPF7.5, without affecting its NO₃⁻ transport activity (Fig. 5E,F; Appendix Fig. S4C,D). There were no differences in either Cl⁻ or NO₃⁻ transport activity between GmNPF7.5^4A and GmNPF7.5 (Fig. 5E,F; Appendix Fig. S4C,D). Interestingly, GmPI4Kγ4 did not affect the Cl⁻ transport activity of GmNPF7.5^4A (Fig. 5E and Appendix Fig. S4C), suggesting that phosphorylation on these residues was essential to the GmPI4Kγ4-mediated inhibition of GmNPF7.5. Furthermore, we constructed a GmNPF7.5 variant with a mutation mimicking phosphorylation at the four residues (GmNPF7.5^S295D/T296D/S336D/T337D, abbreviated GmNPF7.5^4D), and found that GmNPF7.5^4D showed reduced Cl⁻ transport activity compared to GmNPF7.5, whereas their NO₃⁻ transport activity remained unchanged (Fig. EV4). In addition, the co-expression of GmPI4Kγ4 and GmNPF7.5^HapB in oocytes had no effect on the Cl⁻ and NO₃⁻ transport activity of GmNPF7.5^HapB (Fig. EV5).

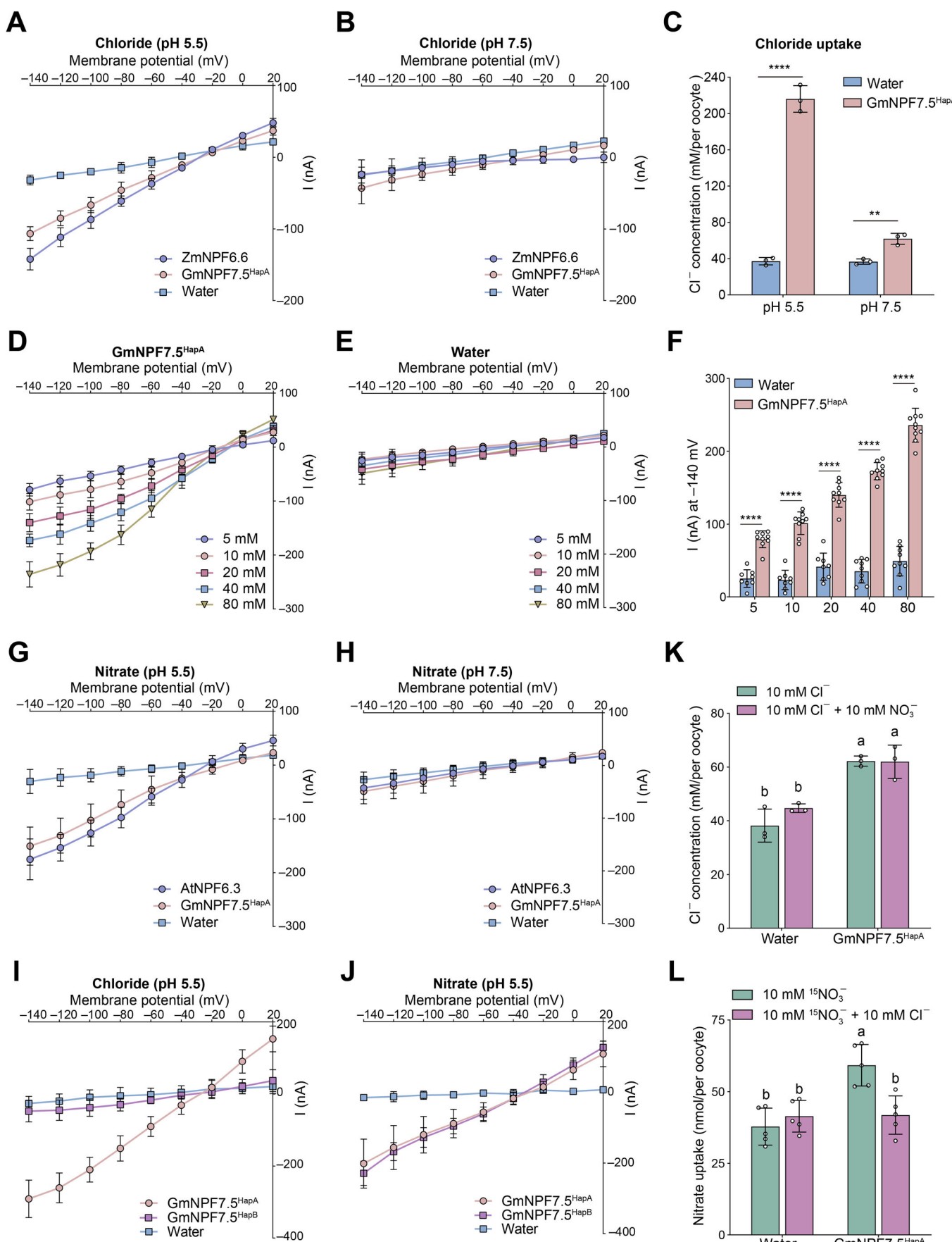

**Figure 3.   GmNPF7.5$^{HapA}$ transported Cl$^-$ and nitrate (NO$_3^-$) in *Xenopus* oocytes but showed a preference for Cl$^-$.**

(A, B) Current–voltage (I–V) relationship for *Xenopus* oocytes expressing ZmNPF6.6 or GmNPF7.5$^{HapA}$. The oocytes were exposed to basal solution containing 10 mM Cl$^-$ at pH 5.5 (A) and pH 7.5 (B) ($n = 8$–10 single oocytes). Oocytes injected with water were used as a background control. (C) Cl$^-$ concentration in oocytes injected with GmNPF7.5$^{HapA}$ cRNA or water. Each replicate ($n = 3$) contained 10 oocytes. Significance was determined using a two-sided Student's *t* test (**$P < 0.01$, ****$P < 0.0001$). *P* values < 0.0001 (pH 5.5), *P* values = 0.0031 (pH 7.5). The experiment was repeated three times, with similar results. (D, E) Cl$^-$-elicited currents mediated by GmNPF7.5$^{HapA}$ were dependent on external Cl$^-$ concentrations. *Xenopus* oocytes injected with *GmNPF7.5* cRNA (D) or water (E) were exposed to basal solution containing 5–80 mM Cl$^-$ at pH 5.5. (F) Comparison of currents elicited by –140 mV between oocytes expressing GmNPF7.5$^{HapA}$ and a water-injected control ($n = 8$–10 single oocytes). Significance was determined using a two-sided Student's *t* test (****$P < 0.0001$). *P* values < 0.0001 (5 mM), <0.0001 (10 mM), <0.0001 (20 mM), <0.0001 (40 mM), <0.0001 (80 mM). (G, H) I–V relationship for *Xenopus* oocytes expressing AtNPF6.3 or GmNPF7.5$^{HapA}$. Oocytes were exposed to basal solution containing 10 mM NO$_3^-$ at pH 5.5 (G) and pH 7.5 (H) ($n = 8$–10 single oocytes). (I, J) I–V relationship for *Xenopus* oocytes expressing the two haplotypes of GmNPF7.5 (GmNPF7.5$^{HapA}$ or GmNPF7.5$^{HapB}$). Oocytes were exposed to basal solution containing 10 mM Cl$^-$ (I) or NO$_3^-$ (J) at pH 5.5 ($n = 6$–9 single oocytes). (K) Cl$^-$ concentrations in oocytes exposed to 10 mM Cl$^-$ in the presence or absence of an equal concentration of NO$_3^-$. Data are means ± SEM ($n = 3$; each replicate contained 10 oocytes). Significance was determined using two-way ANOVA, followed by Tukey's test. Different letters indicate significant differences ($P < 0.05$). *P* values = 0.5273 (Water-10 mM Cl$^-$ vs Water-10 mM Cl$^-$ + 10 mM NO$_3^-$), 0.0012 (Water-10 mM Cl$^-$ vs GmNPF7.5$^{HapA}$-10 mM Cl$^-$), 0.0012 (Water-10 mM Cl$^-$ vs GmNPF7.5$^{HapA}$-10 mM Cl$^-$ + 10 mM NO$_3^-$), 0.009 (Water-10 mM Cl$^-$ + 10 mM NO$_3^-$ vs GmNPF7.5$^{HapA}$-10 mM Cl$^-$), 0.0098 (Water-10 mM Cl$^-$ + 10 mM NO$_3^-$ vs GmNPF7.5$^{HapA}$-10 mM Cl$^-$ + 10 mM NO$_3^-$), >0.9999 (GmNPF7.5$^{HapA}$-10 mM Cl$^-$ vs GmNPF7.5$^{HapA}$-10 mM Cl$^-$ + 10 mM NO$_3^-$). (L) NO$_3^-$ uptake amounts in oocytes exposed to 10 mM $^{15}$NO$_3^-$ in the presence or absence of an equal concentration of Cl$^-$. Data are means ± SEM ($n = 5$, each replicate contained 2 oocytes). Significance was determined using two-way ANOVA, followed by Tukey's test. Different letters indicate significant differences ($P < 0.05$). *P* values = 0.8149 (Water-10 mM $^{15}$NO$_3^-$ vs Water-$^{15}$NO$_3^-$ + 10 mM Cl$^-$), 0.0005 (Water-10 mM $^{15}$NO$_3^-$ vs GmNPF7.5$^{HapA}$-10 mM $^{15}$NO$_3^-$), 0.7635 (Water-10 mM $^{15}$NO$_3^-$ vs GmNPF7.5$^{HapA}$-10 mM $^{15}$NO$_3^-$ + 10 mM Cl$^-$), 0.0026 (Water-$^{15}$NO$_3^-$ + 10 mM Cl$^-$ vs GmNPF7.5$^{HapA}$-10 mM $^{15}$NO$_3^-$), 0.9996 (Water-$^{15}$NO$_3^-$ + 10 mM Cl$^-$ vs GmNPF7.5$^{HapA}$-10 mM $^{15}$NO$_3^-$ + 10 mM Cl$^-$), 0.0032 (GmNPF7.5$^{HapA}$-10 mM $^{15}$NO$_3^-$ vs GmNPF7.5$^{HapA}$-10 mM Cl$^-$). Source data are available online for this figure.

Next, we transiently overexpressed *GmNPF7.5* or *GmNPF7.5$^{4A}$* together with *GmPI4Kγ4* in soybean plants (Appendix Fig. S4E). Both *GmNPF7.5*-OE and *GmNPF7.5$^{4A}$*-OE plants displayed similar decreases in plant height and fresh weight upon salt stress (Fig. 5G–I). However, *GmPI4Kγ4* overexpression in *GmNPF7.5*-OE plants rescued the salt-sensitive phenotype, but had no effect on *GmNPF7.5$^{4A}$*-OE plants (Fig. 5G–I). Furthermore, GmPI4Kγ4 overexpression in GmNPF7.5-OE plants alleviated GmNPF7.5 overexpression-induced Cl$^-$ toxicity, as demonstrated by reduced Cl$^-$ accumulation and increased NO$_3^-$ levels. However, these effects vanished when GmPI4Kγ4 was overexpressed in *GmNPF7.5$^{4A}$*-OE plants (Fig. 5J–M), and similar results were observed in the xylem of *GmNPF7.5$^{4A}$*-OE plants under salt stress (Appendix Fig. S5A,B). Together, these results indicate that GmPI4Kγ4 inhibits GmNPF7.5-mediated root Cl$^-$ uptake by direct phosphorylation.

### *GmNPF7.5* is the dominant gene affecting soybean salt tolerance

To further confirm the role of GmNPF7.5 in soybean salt tolerance, we generated a stable transgenic *GmNPF7.5* overexpression line (GmNPF7.5$^{OX}$) and a mutant line based on CRISPR-Cas9 technology (GmNPF7.5$^{crispr}$) (Appendix Fig. S6). There were no significant growth differences between these lines under control conditions. When exposed to salt stress, *GmNPF7.5$^{OX}$* plants produced hypersensitive phenotypes with decreased fresh weight, whereas *GmNPF7.5$^{crispr}$* plants were more tolerant than the WT line (Fig. 6A–D). *GmNPF7.5$^{OX}$* plants showed higher Cl$^-$ accumulation and lower NO$_3^-$ content in shoots, whereas *GmNPF7.5$^{crispr}$* plants showed opposite changes compared to WT upon salt stress (Fig. 6E,F; Appendix Fig. S7). Next, we assessed the grain yield of transgenic *GmNPF7.5* lines under normal and salt stress conditions. Under normal conditions, there were no significant differences in grain yield per plant among lines. However, under salt stress conditions, grain yield per plant was higher in *GmNPF7.5$^{crispr}$* plants and lower in *GmNPF7.5$^{OX}$* plants than in WT plants (Fig. 6G–I). To evaluate the application prospect of GmNPF7.5 in soybean salt

tolerance improvement, we knocked down *GmNPF7.5* expression in five high-yield commercial soybean cultivars ("Dengke 5," "Mengdou 1137," "Hedou 33," "Heinong 84," and "Suinong 52"). As shown in Appendix Fig. S8, the suppression of *GmNPF7.5* expression significantly enhanced salt tolerance and decreased Cl$^-$ uptake in all five cultivars. This discovery establishes the regulatory mechanism underlying Cl$^-$ transport and salt stress tolerance, and introduces a new breeding strategy to develop salt stress-tolerant soybean plants, offering a promising solution to the continually increasing global demand for soybean protein and oil.

## Discussion

In plants, Cl$^-$ is an essential nutritional element that plays roles in osmotic regulation, charge balance, and water photolysis during photosynthesis (Raven, 2017; Wege et al, 2017). However, excessive Cl$^-$ uptake in roots and its accumulation in shoots are toxic to plants, particularly under salt stress (Geilfus, 2018; Li et al, 2017). Previous studies have demonstrated that the maintenance of Cl$^-$ homeostasis is vital for salt tolerance in plants (Li et al, 2017). However, the molecular mechanism for this role has remained elusive. We conducted GWAS and transcriptome analysis to explore the key genetic locus affecting Cl$^-$ homeostasis, and identified the SNP variation GmNPF7.5, which was associated with soybean Cl$^-$ tolerance (Fig. 1). Through phenotypic analysis of soybean plants under NaCl, KCl, or Na$_2$SO$_4$ treatment (Fig. 2A,B), we found that Cl$^-$ stress, rather than Na$^+$ supply, was responsible for NaCl-induced phenotypes among different GmNPF7.5 genotypes, suggesting that GmNPF7.5 determines soybean salt tolerance by mediating Cl$^-$ transport. NaCl-induced gene expression in the root cortex or epidermis and Cl$^-$ contents in tissues and xylem sap indicated that GmNPF7.5 mediated root Cl$^-$ uptake and determined plant Cl$^-$ content under NaCl stress (Figs. 1E, 2C,D, 6E,F and EV2A). Although Cl$^-$ loading into the xylem is considered the critical step controlling plant tolerance to Cl$^-$ stress (Christoph, 2018; Li et al, 2017), root Cl$^-$ uptake should also be crucial to Cl$^-$

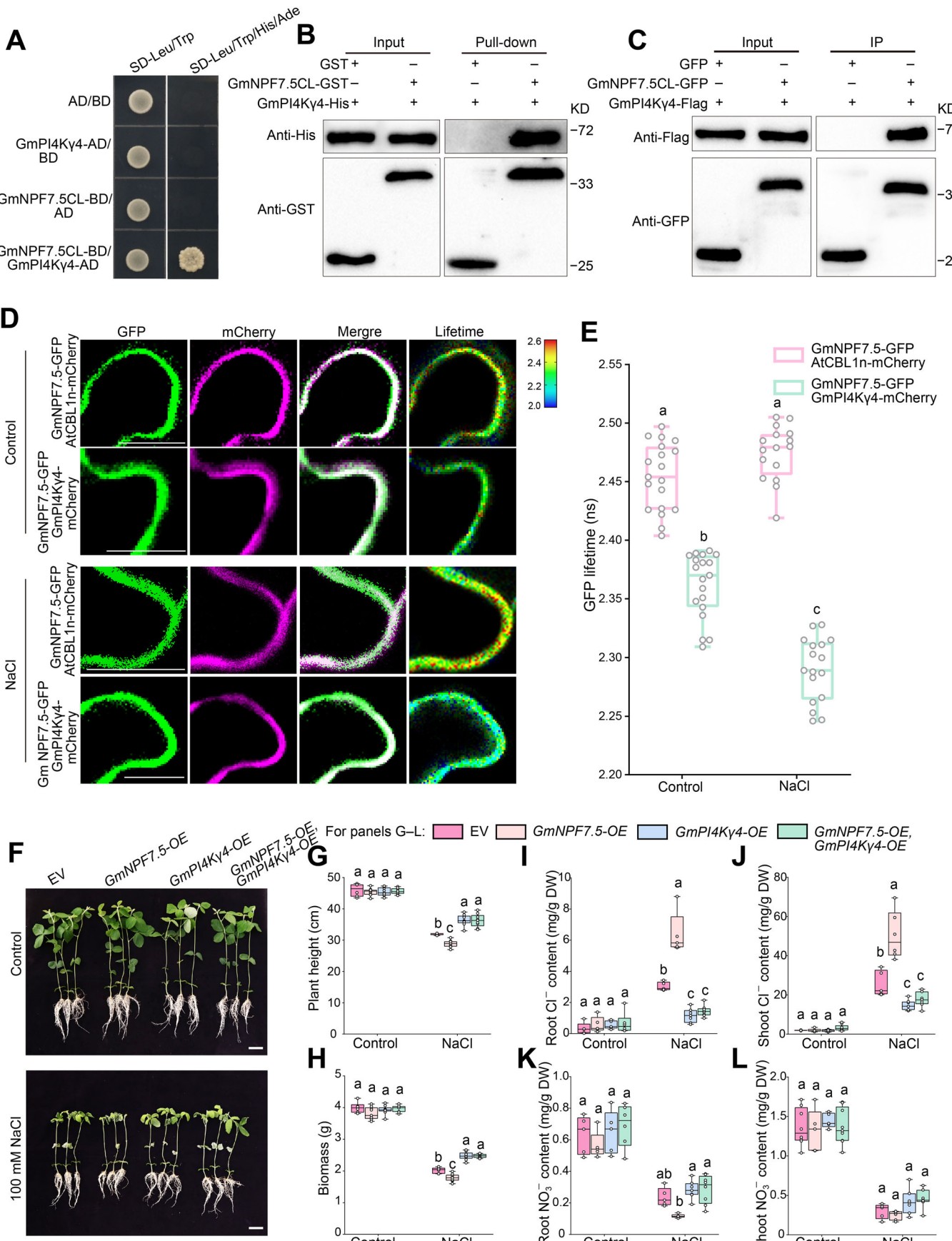

**Figure 4.   GmPI4Kγ4 interacted with GmNPF7.5 and improved salt tolerance by reducing GmNPF7.5-induced Cl⁻ accumulation.**

(A) GmPI4Kγ4 interacted with the central linker domain of GmNPF7.5 (GmNPF7.5CL) in a yeast two-hybrid system. The experiment was repeated three times, with similar results. (B) Results of a pull-down assay based on glutathione-S-transferase (GST) and His tagging, with purified GmNPF7.5CL–GST and GmPI4Kγ4–His. Purified GST was used as a negative control. (C) Coimmunoprecipitation (Co-IP) assay results based on green fluorescent protein (GFP) and flag tagging, showing an interaction between GmNPF7.5CL–GFP and GmPI4Kγ4–flag expressed in *Nicotiana benthamiana* leaves. GFP was used as a negative control. (D, E) Results of a Förster resonance energy transfer by fluorescence lifetime imaging (FRET–FLIM) assay based on co-expression of GmNPF7.5–GFP and GmPI4Kγ4–mCherry in *N. benthamiana* leaves. Scale bars, 20 μm. Results are confocal images (D) and measurements of GFP lifetime (E). The plasma membrane marker AtCBL1n–mCherry was used as a control. Data are means ± SEM ($n > 15$). Significance was determined using two-way ANOVA, followed by Tukey's test. Different letters indicate significant differences ($P < 0.05$). *P* values < 0.0001 (Control-GmNPF7.5-GFP/AtCBL1n-mCherry vs Control-GmNPF7.5-GFP/GmPI4Kγ4-mCherry), 0.1187 (Control-GmNPF7.5-GFP/AtCBL1n-mCherry vs NaCl-GmNPF7.5-GFP/AtCBL1n-mCherry), <0.0001 (Control-GmNPF7.5-GFP/AtCBL1n-mCherry vs NaCl-GmNPF7.5-GFP/GmPI4Kγ4-mCherry), <0.0001 (Control-GmNPF7.5-GFP/GmPI4Kγ4-mCherry vs NaCl-GmNPF7.5-GFP/AtCBL1n-mCherry), <0.0001 (Control-GmNPF7.5-GFP/GmPI4Kγ4-mCherry vs NaCl-GmNPF7.5-GFP/GmPI4Kγ4-mCherry), <0.0001 (NaCl-GmNPF7.5-GFP/AtCBL1n-mCherry vs NaCl-GmNPF7.5-GFP/GmPI4Kγ4-mCherry). (F–H) Phenotype (F), plant height (G), and biomass (H) of soybean plants ('Williams 82') with transgenic hairy roots harboring both or either *GmNPF7.5*-OE and *GmPI4Kγ4*-OE. Scale bars, 5 cm. Data in (G, H) are means ± SEM ($n = 6$–9 individual seedlings). Significance was determined using one-way ANOVA, followed by Tukey's test. Different letters indicate significant differences ($P < 0.05$). G, Control: *P* values = 0.8464 (EV vs *GmNPF7.5*-OE), 0.8532 (EV vs *GmPI4Kγ4*-OE), 0.9444 (EV vs *GmNPF7.5*-OE/*GmPI4Kγ4*-OE), >0.9999 (*GmNPF7.5*-OE vs *GmPI4Kγ4*-OE), 0.9913 (*GmNPF7.5*-OE vs *GmNPF7.5*-OE/*GmPI4Kγ4*-OE), 0.9927 (*GmPI4Kγ4*-OE vs *GmNPF7.5*-OE/*GmPI4Kγ4*-OE). (The following is the same order). NaCl: *P* values = 0.0011, <0.0001, <0.0001, <0.0001, <0.0001, 0.9939. H, Control: *P* values = 0.059, 0.7435, 0.9328, 0.3083, 0.1318, 0.9666. NaCl: *P* values = 0.0061, <0.0001, <0.0001, <0.0001, <0.0001, 0.9996. (I–L) Root Cl⁻ content (I), shoot Cl⁻ content (J), root NO₃⁻ content (K), and shoot NO₃⁻ content (L) of soybean plants. Plants harboring empty vector (EV) were used as a background control. Data in (I–L) are means ± SEM ($n = 5$–9 independent biological replicates). Significance was determined using one-way ANOVA, followed by Tukey's test. Different letters indicate significant differences ($P < 0.05$). (I) Control: *P* values = 0.9142 (EV vs *GmNPF7.5*-OE), 0.9258 (EV vs *GmPI4Kγ4*-OE), 0.7998 (EV vs *GmNPF7.5*-OE/*GmPI4Kγ4*-OE), >0.9999 (*GmNPF7.5*-OE vs *GmPI4Kγ4*-OE), 0.9957 (*GmNPF7.5*-OE vs *GmNPF7.5*-OE/*GmPI4Kγ4*-OE), 0.9934 (*GmPI4Kγ4*-OE vs *GmNPF7.5*-OE/*GmPI4Kγ4*-OE). (The following is the same order). NaCl: *P* values < 0.0001, <0.0001, 0.0003, <0.0001, <0.0001, 0.8024. (J), Control: *P* values > 0.9999, >0.9999, 0.9627, >0.9999, 0.9685, 0.9625. NaCl: *P* values <0.0001, 0.0042, 0.047, <0.0001, <0.0001, 0.7463. (K) Control: *P* values = 0.6829, 0.9927, 0.7339, 0.5145, 0.1437, 0.8791. NaCl: *P* values = 0.2155, 0.8429, 0.7899, 0.0262, 0.0174, 0.9997. (L) Control: *P* values = 0.9741, 0.8475, 0.9998, 0.6819, 0.9582, 0.8705. NaCl: *P* values = 0.9634, 0.6774, 0.4664, 0.3344, 0.1811, 0.9817. Data in (E, G–L) are plotted with box–whisker plots: the whiskers represent maximum and minimum values, and boxes represent the upper quartile, median, and lower quartile, dots represent data points. Source data are available online for this figure.

homeostasis in plants, and the key transporters mediating Cl⁻ uptake from saline soil have not been previously identified. Therefore, the GmNPF7.5 identified in this study may fill a gap in knowledge about plant Cl⁻ transport.

GmNPF7.5 mediated both NO₃⁻ and Cl⁻ uptake but was more selective for Cl⁻, when expressed in *Xenopus* oocytes (Figs. 3 and EV3C,D). Notably, GmNPF7.5 expression was specifically induced by high Cl⁻ stress (Fig. EV1B). The Cl⁻-induced expression of *GmNPF7.5* may greatly promote Cl⁻ uptake and accumulation, which appears to be a self-destructive or "suicidal" behavior. However, we discovered that NaCl stress also enhanced *GmPI4Kγ4* expression and GmPI4Kγ4–GmNPF7.5 interaction (Fig. 4D,E; Appendix Fig. S2C). GmPI4Kγ4 phosphorylated GmNPF7.5 and inhibited its Cl⁻ uptake, protecting soybean plants from Cl⁻ toxicity. Because GmPI4Kγ4 had no effect on the NO₃⁻ permeability of GmNPF7.5, Cl⁻-induced *GmNPF7.5* expression appears to facilitate NO₃⁻ uptake under salt stress, improving salt tolerance (Fig. 5; Appendix Fig. S4). Therefore, Cl⁻-induced *GmNPF7.5* expression can be considered a blessing in disguise, rather than suicidal behavior. In this mechanism, GmPI4Kγ4 acts as a molecular switch to protect soybean plants from GmNPF7.5-triggered Cl⁻ toxicity under salt stress. MtNPF6.5, an NPF protein from *Medicago*, exhibited similar Cl⁻ selectivity with GmNPF7.5; however, its expression is downregulated by NaCl stress (Xiao et al, 2021), resulting in the inhibition of root Cl⁻ uptake under NaCl stress. The expression of these two NPFs differs greatly in response to Cl⁻ stress, indicating that *Medicago* and *G. max* employ different strategies to cope with Cl⁻-selective NPFs under Cl⁻ stress.

The phosphorylation sites of GmNPF7.5 by GmPI4Kγ4 were located in the intracellular linker domain (Fig. 5C). According to the structure of AtNPF6.3, the N-terminal region of the linker can form an amphipathic α-helix, providing a potential protein docking site (Sun et al, 2014), which may recruit interacting proteins such as

GmPI4Kγ4, identified in this study. However, the mechanism by which phosphorylation affects the Cl⁻ permeability of GmNPF7.5 remains unknown, and should be elucidated in further studies through the identification of substrate binding sites and structural analysis of GmNPF7.5. Similar to the phosphorylation modification in the intracellular linker, an SNP1735 variation-induced V579I mutation in the C-terminal cytosolic domain also inhibited Cl⁻ uptake of GmNPF7.5, without affecting NO₃⁻ permeability (Fig. 3I,J). However, the C-terminal cytosolic domain of GmNPF7.5 is very short, comprising only 22 amino acid residues, and there is little information regarding its function and structure. Therefore, the mechanisms by which the V579I variation affects Cl⁻ permeability and whether there are interaction effects between the C-terminal SNP1735 site and the phosphorylation modification in the central linker also remain unknown. Despite these unresolved questions, our findings indicate that these two intracellular domains play crucial roles in regulating the Cl⁻ transport activity of NPF transporters.

We found that GmNPF7.5 is a key gene determining soybean Cl⁻ tolerance under salt stress, providing an effective strategy for breeding salt-tolerant soybean varieties. The use of CRISPR transgenic lines and manipulation of *GmNPF7.5* expression in high-yield commercial cultivars showed promising results (Fig. 6A–I; Appendix Figs. S7 and 8). Besides NaCl stress, excessive application of fertilizers containing Cl⁻, such as KCl or NH₄Cl, may also cause Cl⁻ stress. Therefore, GmNPF7.5 may also represent a promising target for improving Cl⁻ tolerance under overfertilization. In conclusion, we identified an SNP variation, *GmNPF7.5*, associated with shoot Cl⁻ content, and discovered a novel mechanism for maintaining Cl⁻ homeostasis and improving soybean plant salt tolerance (Fig. 6J). GmNPF7.5 significantly attenuated Cl⁻ permeability but sustained NO₃⁻ uptake, improving salt tolerance in cultivars with this haplotype. Salt stress-induced *GmNPF7.5* expression, resulting in excessive Cl⁻ uptake and

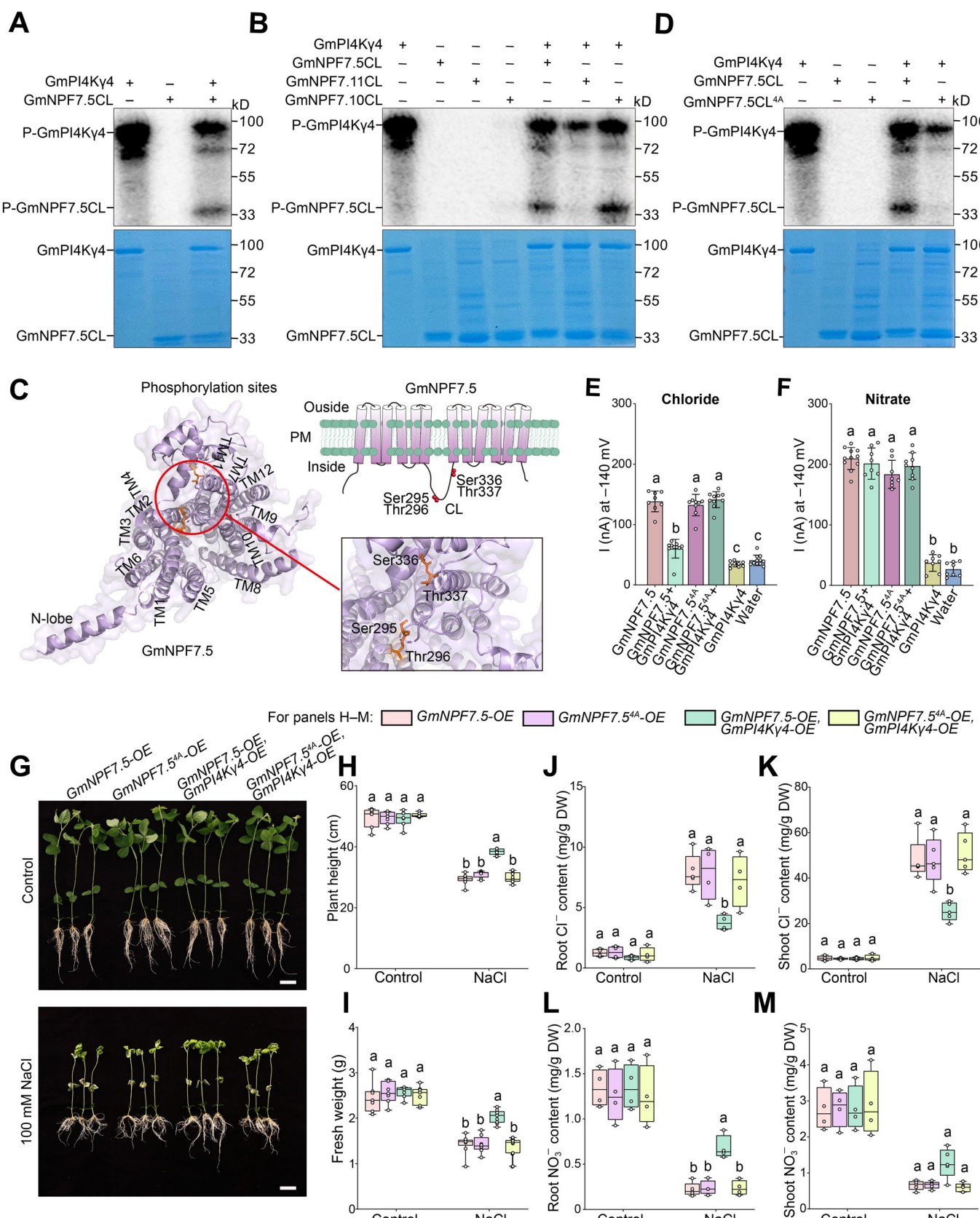

◀ **Figure 5.  GmNPF7.5 was phosphorylated and inhibited by GmPI4Kγ4, and phosphorylation was essential for GmNPF7.5 inactivation and soybean salt tolerance.**

(A) In vitro kinase assay results showing that the central linker domain of GmNPF7.5 (GmNPF7.5CL) was phosphorylated by GmPI4Kγ4. Phosphorylation was examined by autoradiography (top). Input proteins were stained with Coomassie Brilliant Blue (CBB, bottom). All proteins used in the kinase activity assay were fused with the GST tag. (B) GmPI4Kγ4 phosphorylated GmNPF7.5 and GmNPF7.10, but not GmNPF7.11. All proteins used in the kinase activity assay were fused with the GST tag. (C) Protein structure of GmNPF7.5 and locations of phosphorylation sites. (D) Identification of phosphorylation sites of GmNPF7.5 by GmPI4Kγ4. GmNPF7.5CL$^{4A}$ represented a variant with the mutations S295A/T296A/S336A/T337A. All proteins used in the kinase activity assay were fused with the GST tag. (E) Cl⁻-elicited currents at –140 mV recorded from *Xenopus* oocytes expressing GmNPF7.5, GmNPF7.5$^{4A}$, GmPI4Kγ4, GmNPF7.5 + GmPI4Kγ4, or GmNPF7.5$^{4A}$ + GmPI4Kγ4 in basal solution containing 10 mM Cl⁻ at pH 5.5. (F) NO$_3^-$-elicited currents at –140 mV recorded from *Xenopus* oocytes expressing GmNPF7.5, GmNPF7.5$^{4A}$, GmPI4Kγ4, GmNPF7.5 + GmPI4Kγ4, or GmNPF7.5$^{4A}$ + GmPI4Kγ4 in basal solution containing 10 mM NO$_3^-$ at pH 5.5. Data in (E, F) are means ± SEM ($n$ = 8–11 single oocytes). Significance was determined using one-way ANOVA, followed by Tukey's test. Different letters indicate significant differences ($P < 0.05$). (E) $P$ values < 0.0001 (GmNPF7.5 vs GmNPF7.5 + GmPI4Kγ4), 0.9358 (GmNPF7.5 vs GmNPF7.5$^{4A}$), 0.9976 (GmNPF7.5 vs GmNPF7.5$^{4A}$ + GmPI4Kγ4), <0.0001 (GmNPF7.5 vs GmPI4Kγ4), <0.0001 (GmNPF7.5 vs Water), <0.0001 (GmNPF7.5 + GmPI4Kγ4 vs GmNPF7.5$^{4A}$), <0.0001 (GmNPF7.5 + GmPI4Kγ4 vs GmNPF7.5$^{4A}$ + GmPI4Kγ4), 0.0017 (GmNPF7.5 + GmPI4Kγ4 vs GmPI4Kγ4), 0.0217 (GmNPF7.5 + GmPI4Kγ4 vs Water), 0.6975 (GmNPF7.5$^{4A}$ vs GmNPF7.5$^{4A}$ + GmPI4Kγ4), <0.0001 (GmNPF7.5$^{4A}$ vs GmPI4Kγ4), <0.0001 (GmNPF7.5$^{4A}$ vs Water), <0.0001 (GmNPF7.5$^{4A}$ + GmPI4Kγ4 vs GmPI4Kγ4), <0.0001 (GmNPF7.5$^{4A}$ + GmPI4Kγ4 vs Water), 0.9007 (GmPI4Kγ4 vs Water). (The following is the same order). (F) $P$ values = 0.9413, 0.0683, 0.7267, <0.0001, <0.0001, 0.4796, 0.9982, <0.0001, <0.0001, 0.7129, <0.0001, <0.0001, <0.0001, <0.0001, and 0.8992. (G–I) Phenotypes (G), plant heights (H), and biomass (I) of soybean plants ('Williams 82') with transgenic hairy roots harboring the indicated constructs. Scale bar, 5 cm. Data in (H, I) are means ± SEM ($n$ = 7–9 individual seedlings). Significance was determined using one-way ANOVA, followed by Tukey's test. Different letters indicate significant differences ($P < 0.05$). The experiment was repeated three times, with similar results. (H) Control: $P$ values = 0.9986 (*GmNPF7.5*-OE vs *GmNPF7.5$^{4A}$*-OE), 0.9825 (*GmNPF7.5*-OE vs *GmNPF7.5*-OE + *GmPI4Kγ4*-OE), 0.886 (*GmNPF7.5*-OE vs *GmNPF7.5$^{4A}$*-OE + *GmPI4Kγ4*-OE), 0.9965 (*GmNPF7.5$^{4A}$*-OE vs *GmNPF7.5*-OE + *GmPI4Kγ4*-OE), 0.8134 (*GmNPF7.5$^{4A}$*-OE vs *GmNPF7.5$^{4A}$*-OE + *GmPI4Kγ4*-OE), 0.6927 (*GmNPF7.5*-OE + *GmPI4Kγ4*-OE vs *GmNPF7.5$^{4A}$*-OE + *GmPI4Kγ4*-OE). (The following is the same order). NaCl: $P$ values = 0.2997, <0.0001, 0.9581, <0.0001, 0.5793 and <0.0001. (I), Control: $P$ values = 0.8043, 0.7324, 0.9604, 0.9992, 0.9768 and 0.9494. NaCl: $P$ values = 0.9995, <0.0001, 0.9451, <0.0001, 0.9737, and <0.0001. (J–M) Root Cl⁻ content (J), shoot Cl⁻ content (K), root NO$_3^-$ content (L), and shoot NO$_3^-$ content (M) of soybean plants. Data in (J–M) are means ± SEM ($n$ = 4–5 biological independent samples). Significance was determined using one-way ANOVA, followed by Tukey's test. Different letters indicate significant differences ($P < 0.05$). (J) Control: $P$ values > 0.9999 (*GmNPF7.5*-OE vs *GmNPF7.5$^{4A}$*-OE), 0.9727 (*GmNPF7.5*-OE vs *GmNPF7.5*-OE + *GmPI4Kγ4*-OE), 0.9982 (*GmNPF7.5*-OE vs *GmNPF7.5$^{4A}$*-OE + *GmPI4Kγ4*-OE), 0.9648 (*GmNPF7.5$^{4A}$*-OE vs *GmNPF7.5*-OE + *GmPI4Kγ4*-OE), 0.9967 (*GmNPF7.5$^{4A}$*-OE vs *GmNPF7.5$^{4A}$*-OE + *GmPI4Kγ4*-OE), 0.9936 (*GmNPF7.5*-OE + *GmPI4Kγ4*-OE vs *GmNPF7.5$^{4A}$*-OE + *GmPI4Kγ4*-OE). (The following is the same order). NaCl: $P$ values = 0.9998, 0.0002, 0.8076, 0.0006, 0.8691 and 0.0038. (K), Control: $P$ values > 0.9999, >0.9999, >0.9999, >0.9999 and 0.9999. NaCl: $P$ values = 0.9997, <0.0001, 0.8858, <0.0001, 0.8451, and <0.0001. (L) Control: $P$ values = 0.9454, 0.9999, 0.9225, 0.9234, 0.9998, and 0.8961. NaCl: $P$ values = 0.999, 0.0114, 0.9993, 0.0232, >0.9999 and 0.0224. (M) Control: $P$ values = 0.9995, 0.9998, 0.9782, >0.9999, 0.9917, and 0.9885. NaCl: $P$ values > 0.9999, 0.1897, 0.996, 0.1986, 0.9944 and 0.1252. Data in (H–M) are plotted with box–whisker plots: the whiskers represent maximum and minimum values, and boxes represent the upper quartile, median, and lower quartile, dots represent data points. Source data are available online for this figure.

toxicity. To relieve Cl⁻ toxicity, *GmPI4Kγ4* expression was also induced under salt stress. GmPI4Kγ4 phosphorylated GmNPF7.5 and inhibited its Cl⁻ uptake, resulting in decreased Cl⁻ accumulation and lower toxicity. Together, these findings offer a new regulatory mechanism for the improvement of salt tolerance in soybean plants, providing an important route for breeding salt-tolerant cultivars.

# Methods

### Reagents and tools table

| Reagent/resource | Reference or source | Identifier or catalog number |
|---|---|---|
| **Antibodies** | | |
| GST-tag Antibody [HRP] | GenScript | A00130 |
| His-tag Antibody, Mouse | GenScript | A00186 |
| Mouse anti GFP-tag | ABclonal | AE012 |
| Mouse anti DDDDK-Tag | ABclonal | AE005 |
| **Oligonucleotides and other sequence-based reagents** | | |
| PCR and qRT-PCR primers | This study | Table EV 1 |
| **Chemicals, enzymes, and other reagents** | | |
| NaCl | Sigma-Aldrich | 31434 |
| NaOH | Sigma-Aldrich | 221465 |
| Salicylic acid | Sigma-Aldrich | W398500 |
| Agarose | Sigma-Aldrich | A0576 |

| Reagent/resource | Reference or source | Identifier or catalog number |
|---|---|---|
| KCl | Sigma-Aldrich | P9541 |
| Na$_2$SO$_4$ | Sigma-Aldrich | 238597 |
| MgCl$_2$ | Sigma-Aldrich | M2393 |
| CaCl$_2$ | Sigma-Aldrich | C3306 |
| HEPES | Sigma-Aldrich | H3375 |
| MES | Sigma-Aldrich | 475893 |
| Tetracycline | Sigma-Aldrich | 58346-M |
| Streptomycin sulfate | Sigma-Aldrich | 5711 |
| Bis-Tris propane | Sigma-Aldrich | 64431-96-5 |
| Na$^{15}$NO$_3$ | Sigma-Aldrich | 364606 |
| NaNO$_3$ | Sigma-Aldrich | S8170 |
| SD-Leu/Trp | Coolaber | PM2221 |
| SD-Leu/Trp/His/Ade | Coolaber | PM2112 |
| Tris-HCl | Sigma-Aldrich | 10812846001 |
| EDTA | Sigma-Aldrich | 4005-OP |
| Dimethyl sulfoxide (DMSO) | Sigma-Aldrich | D8418 |
| D-Mannitol | Sigma-Aldrich | 1371621000 |
| PEG3350 | Sigma-Aldrich | P4338 |
| PBS | Sigma-Aldrich | P3813 |
| T7 RiboMAX large-scale RNA production system | Promega | P1300 |
| Ribo m7G Cap Analog | Promega | P1711 |

| Reagent/resource | Reference or source | Identifier or catalog number |
|---|---|---|
| NheI | Thermo Fisher | FD0947 |
| SphI | Thermo Fisher | FD0604 |
| RNA isolater Total RNA Extraction Reagent | Vazyme | R401-01 |
| HiScript II Q RT SuperMix for qPCR | Vazyme | R223-01 |
| Hieff qPCR SYBR Green Master Mix | YESEAN | 11201ES |
| Salmon Sperm DNA Solution | Thermo Fisher | 15632011 |
| Glutathione Resin | GenScript | L00206 |
| **Software** | | |
| GraphPad Prism 8.0 | https://www.graphpad.com | |
| ImageJ | https://imagej.nih.gov/ij/index.html | |
| **Other** | | |
| Hiseq 2500 platform | Illumina | |

## Plant materials and growth conditions

For the GWAS, we obtained 198 soybean accessions (Zhang et al, 2021) from Jiangsu Academy of Agricultural Sciences, and grew them in a culture room at 25 °C under long-day (16 h light/8 h dark) conditions. The seedlings were grown for 2 weeks under normal condition before treatment with 150 mM NaCl. Shoot tissues were collected for $Cl^-$ content measurements.

## Measurement of $Cl^-$ and $NO_3^-$ concentrations

Prior to sample pretreatment, shoot or root samples were cleaned and then dried at 105 °C until constant weight was achieved (~30 min). Weighed the dry samples, ground it in a 2 mL centrifuge tube, added 1 mL water, and boiled it for 30 min. Then the samples were centrifuged for 15 min at $13,000 \times g$.

$Cl^-$ concentrations were measured following a previously described method (Xiao et al, 2021), with modifications. The supernatant was filtered through 0.22-μm mesh, and $Cl^-$ content was determined using a chromatograph (ICS 1100, Thermo Fisher Scientific, Waltham, MA, USA).

$NO_3^-$ concentrations were analyzed following a previously described method (Liu et al, 2023), with modifications. The 20 μL sample solution was absorbed into a 2 mL centrifuge tube, and 80 μL 5% sulfuric acid–salicylic acid solution was added. After mixing, the solution was reacted for 20 min in the dark, and then 900 μL 8% NaOH solution was added, followed by mixing through inversion until the sample had cooled to room temperature. Finally, the optical density at 410 nm ($OD_{410}$) was measured.

## $Cl^-$ content determination for GWAS

$Cl^-$ content was determined in 198 soybean accessions that had previously been sequenced and subjected to population structure

and linkage disequilibrium analyses (Zhang et al, 2021). We identified 1,802,144 SNPs using the Genome-wide Complex Trait Analysis (GCTA) software in the PLINK platform, based on a minor allele frequency ≥0.05, missingness rate <0.05, and pairwise independence assessed using a window of 50 kb, step size of 10, and $r^2$ threshold of 0.2 (Purcell et al, 2007; Yang et al, 2011). Ultimately, we obtained 41,125 independent SNPs. The GWAS mixed linear model was run using the FaST-LMM program (Lippert et al, 2011), with the threshold for significant association set to $1/n$, where $n$ is the effective number of independent SNPs ($P < 2.43 \times 10^{-5}$ or $-\log_{10}(P) > 4.6$).

## Haplotype analysis of *GmNPF7.5*

For haplotype analysis of *GmNPF7.5*, the 198 accessions were divided into two haplotypes based on three SNPs (SNP257, SNP786, and SNP1735) in the *GmNPF7.5* exon. Shoot $Cl^-$ content was compared between the haplotypes using the Student's $t$ test. If not specifically stated, the GmNPF7.5 represents GmNPF7.5[HapA] in most experiments.

## RNA-seq analysis

RNA-seq analysis was performed using the BMKCloud platform (www.biocloud.net). "Williams 82" seedlings were grown under control conditions for 2 weeks and then treated with 150 mM NaCl for 6 h. The root tissues of the seedlings were collected for RNA extraction. Three biological replicates each were prepared for the control and NaCl treatments, with three seedlings per replicate. RNA-seq libraries were constructed using the NEB Next Ultra II RNA Library Prep Kit for Illumina (New England Biolabs, Ipswich, MA, USA), according to the manufacturer's instructions. Libraries were sequenced on the Hiseq 2500 platform (Illumina, San Diego, CA, USA) for 150 bp paired-end reads. RNA-seq reads were filtered and aligned to the reference genome using the Hisat2 alignment tool (Kim et al, 2015). Only uniquely mapped reads were retained. RNA-seq reads were normalized to fragments per kb of transcript per million mapped reads for each sample. The R package *DESeq2* was used to identify DEGs between the two groups under control and salt stress conditions (Wagner et al, 2012), using the criteria $\log_2(|\text{fold change}|) \geq 1$ and false discovery rate ≤0.01.

## Transformation of hairy roots of soybean

The hairy roots were transformed using *Agrobacterium rhizogenes*, as described previously (Wei et al, 2016). Unless specifically indicated, the recipient plants were "Williams 82" (a cultivar with HapA). To generate the overexpression construct, the coding sequence (CDS) of GmNPF7.5 was amplified and ligated into pBA002 vector. To generate the RNAi construct, a 312 bp fragment was amplified and inserted into pFGC5941 vector. To generate the GmNPF7.5–GmPI4Kγ4 co-expression construct, the CDSs of GmNPF7.5 and GmPI4Kγ4 were ligated into pUB–GFP vector containing pGmUbi:3×flag (Wang et al, 2021a). Then these vectors were introduced into *A. rhizogenes* strain "K599." Briefly, 5- to 7-day-old "Williams 82" soybean seedlings were infected with the "K599" strain, and transferred to humid vermiculite for 1 week. The infected seedlings were transferred to ½ Hoagland nutrient solution after the callus had grown. Newly grown hairy roots were sampled

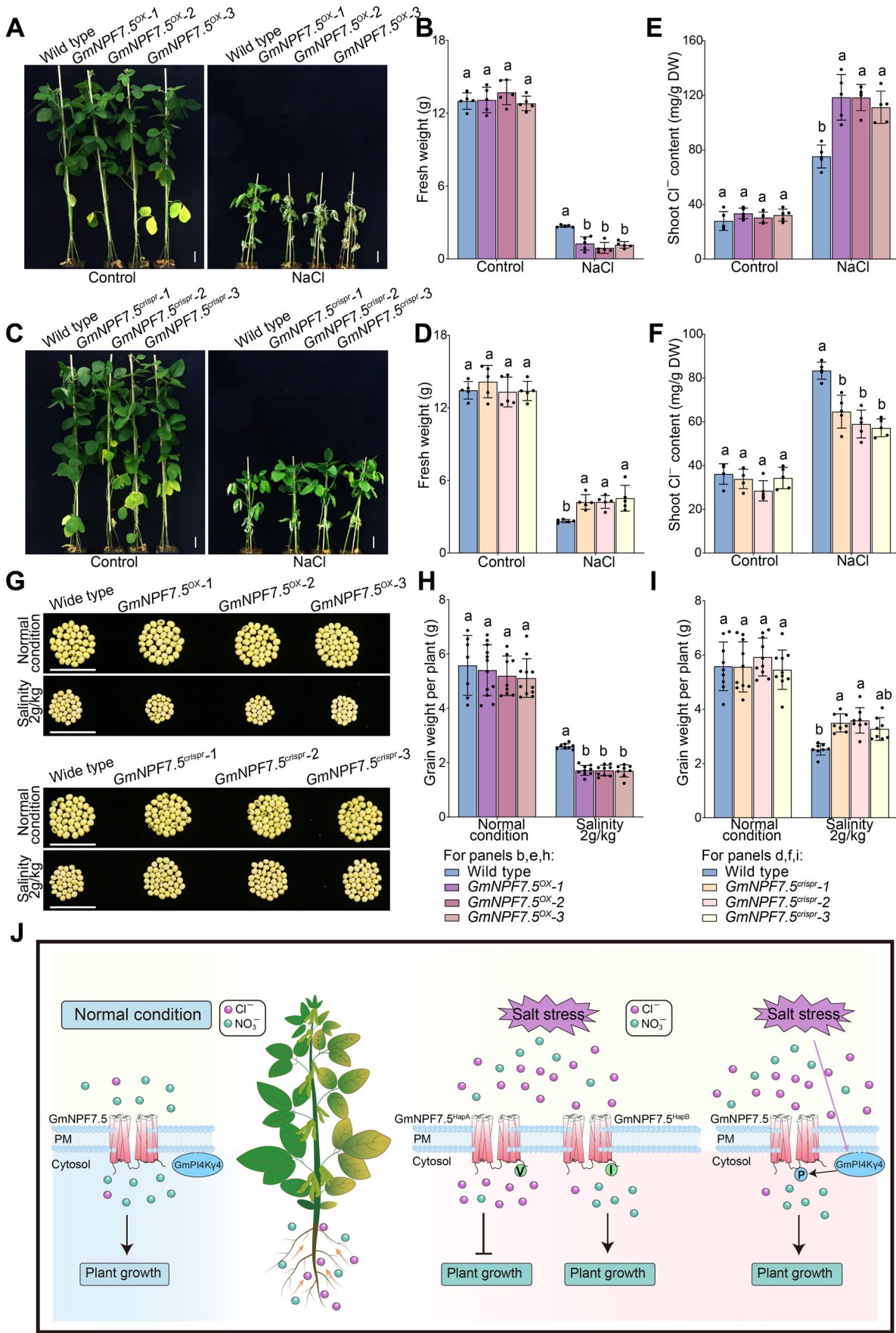

◄

**Figure 6.  GmNPF7.5 negatively regulated salt tolerance in stable transgenic soybean plants.**

(A–F) Phenotypes (A, C), fresh weight (B, D), and shoot Cl⁻ content (E, F) of *GmNPF7.5* transgenic and wild-type (WT) soybean plants under control and salt treatments. Scale bars, 5 cm. Data in (B, D–F) are means ± SEM ($n = 5$ biological independent samples). Significance was determined using one-way ANOVA, followed by Tukey's test. Different letters indicate significant differences ($P < 0.05$). (B), Control: $P$ values = 0.9963 (Wide type vs *GmNPF7.5*$^{OX}$-1), 0.3377 (Wide type vs *GmNPF7.5*$^{OX}$-2), 0.9692 (Wide type vs *GmNPF7.5*$^{OX}$-3), 0.4561 (*GmNPF7.5*$^{OX}$-1 vs *GmNPF7.5*$^{OX}$-2), 0.9088 (*GmNPF7.5*$^{OX}$-1 vs *GmNPF7.5*$^{OX}$-3), 0.1578 (*GmNPF7.5*$^{OX}$-2 vs *GmNPF7.5*$^{OX}$-3). (The following is the same order). NaCl: $P$ values = 0.0097, 0.0009, 0.0046, 0.8202, 0.9915 and 0.9369. (E) Control: $P$ values = 0.7754, 0.9717, 0.8776, 0.9541, 0.9967 and 0.9895. NaCl: $P$ values < 0.0001, <0.0001, >0.9999, 0.5963 and 0.6129. (D) Control: $P$ values = 0.5799 (Wide type vs *GmNPF7.5*$^{cridpr}$-1), 0.9952 (Wide type vs *GmNPF7.5*$^{cridpr}$-2), 0.9997 (Wide type vs *GmNPF7.5*$^{cridpr}$-3), 0.4373 (*GmNPF7.5*$^{cridpr}$-1 vs *GmNPF7.5*$^{cridpr}$-2), 0.5251 (*GmNPF7.5*$^{cridpr}$-1 vs *GmNPF7.5*$^{cridpr}$-3), 0.9988 (*GmNPF7.5*$^{cridpr}$-2 vs *GmNPF7.5*$^{cridpr}$-3). (The following is the same order). NaCl: $P$ values = 0.039, 0.038, 0.0099, >0.9999, 0.9434 and 0.9467. (F) Control: $P$ values = 0.9015, 0.1106, 0.9461, 0.3655, 0.9991 and 0.2973. NaCl: $P$ values < 0.0001, <0.0001, <0.0001, 0.3316, 0.1301, and 0.949. (G) Grain yield per plant for indicated genotypes grown under normal and salt stress conditions (2 g/kg). Scale bars, 5 cm. (H, I) Grain weight per plant. Data in (H, I) are means ± SEM ($n = 7 - 12$ biological independent samples). Significance was determined using one-way ANOVA, followed by Tukey's test. Different letters indicate significant differences ($P < 0.05$). (H) Control: $P$ values = 0.9397 (Wide type vs *GmNPF7.5*$^{OX}$-1), 0.6473 (Wide type vs *GmNPF7.5*$^{OX}$-2), 0.4508 (Wide type vs *GmNPF7.5*$^{OX}$-3), 0.8889 (*GmNPF7.5*$^{OX}$-1 vs *GmNPF7.5*$^{OX}$-2), 0.709 (*GmNPF7.5*$^{OX}$-1 vs *GmNPF7.5*$^{OX}$-3), 0.9916 (*GmNPF7.5*$^{OX}$-2 vs *GmNPF7.5*$^{OX}$-3). (The following is the same order). NaCl: $P$ values = 0.0329, 0.0318, 0.0349, >0.9999, >0.9999, > 0.9999. (I) Control: $P$ values = 0.9999 (Wide type vs *GmNPF7.5*$^{cridpr}$-1), 0.6696 (Wide type vs *GmNPF7.5*$^{cridpr}$-2), 0.9739 (Wide type vs *GmNPF7.5*$^{cridpr}$-3), 0.6039 (*GmNPF7.5*$^{cridpr}$-1 vs *GmNPF7.5*$^{cridpr}$-2), 0.9835 (*GmNPF7.5*$^{cridpr}$-1 vs *GmNPF7.5*$^{cridpr}$-3), 0.3844 (*GmNPF7.5*$^{cridpr}$-2 vs *GmNPF7.5*$^{cridpr}$-3). (The following is the same order). NaCl: $P$ values = 0.0211, 0.0098, 0.1111, 0.9926, 0.9028 and 0.7727. (J) Proposed model of GmPI4Kγ4-mediated regulation of GmNPF7.5 in response to salt stress. Under normal conditions, GmNPF7.5 localized at the plasma membrane mediates the uptake of Cl⁻ and NO₃⁻ as nutrient anions. When soybean plants were subjected to salt (Cl⁻) stress, *GmNPF7.5* expression was induced by excessive Cl⁻. In haplotypes with the Val$^{579}$ residue, GmNPF7.5$^{HapA}$ mediated Cl⁻ uptake and increased Cl⁻ accumulation, resulting in ionic toxicity and salt sensitivity. However, in haplotypes with the Ile$^{579}$ residue, GmNPF7.5$^{HapB}$ lost Cl⁻ permeability, but retained NO₃⁻ transport activity, thereby repressing Cl⁻ accumulation and enhancing salt stress tolerance. To protect soybean plants from excessive Cl⁻ accumulation, the expression of GmPI4Kγ4, which phosphorylates GmNPF7.5 and inhibits its Cl⁻ transport activity, was also activated under salt stress. This mechanism repressed GmNPF7.5-induced Cl⁻ accumulation, thereby enhancing soybean salt stress tolerance. Source data are available online for this figure.

from the callus and identified by PCR; positive roots were treated with NaCl, KCl, or Na₂SO₄ at the 2–3 compound leaf stage.

## Stable transformation of soybean

To generate the GmNPF7.5 overexpression construct, the CDS of GmNPF7.5 was amplified from "Williams 82" seedlings and ligated into pBA002 vector. The construct was introduced into *Agrobacterium tumefaciens* strain "EHA105" and transformed into "Williams 82" using the cotyledon node method (Luth et al, 2015). To knock out GmNPF7.5, two sgRNAs (http://crispr.hzau.edu.cn/CRISPR2/) were generated into CRISPR-Cas9 binary vector, and then the construct was introduced into *A. tumefaciens* strain "EHA105" and transformed into soybean "Williams 82" (Luth et al, 2015).

## qRT-PCR analysis

Total RNA was isolated from soybean roots using an RNA isolator (R401–01, Vazyme Biotech, Nanjing, China), following the manufacturer's instructions. First-strand cDNA was synthesized using HiScript II Q RT Supermix (R223–01, Vazyme) from 1 μg total RNA. We conducted qRT-PCR using Hieff qPCR SYBR Green Master Mix (11201ES, Yeasen Biotechnology, Shanghai, China) under the following thermal conditions: 5 min at 95 °C, followed by 40 cycles including 10 s at 95 °C, 20 s at 60 °C, and 20 s at 72 °C. *GmELF* was used as a reference gene. At least three biological repeats were conducted for each treatment. Relative gene expression levels were analyzed as described previously (Deng et al, 2022). The primers used for qRT-PCR are listed in Appendix Table S1.

## GmNPF7.5 transport assays in *X. laevis* oocytes

The transport function of GmNPF7.5 was analyzed experimentally using *Xenopus* oocytes as described previously (Tian et al, 2021). The CDS sequences of *GmNPF7.5*, *ZmNPF6.6*, *AtNPF6.3*, or

*GmPI4Kγ4*, and sequences of *GmNPF7.5*$^{4A}$ or *GmNPF7.5*$^{4D}$ were cloned into pGEMHE vector. All pGEMHE plasmids were linearized using *NheI* or *SphI*. The cRNAs were transcribed and capped in vitro using the T7 RiboMAX large-scale RNA production system (Promega, Madison, WI, USA). Then 46 nL cRNA or water was injected into *Xenopus* oocytes, which then were incubated in a modified solution containing 96 mM NaCl, 2 mM KCl, 5 mM MgCl₂, 5 mM HEPES, 0.6 mM CaCl₂, 25 mg/L tetracycline, and 100 mg/L streptomycin sulfate (pH 7.6) for 2 days at 18 °C after injection (Wen et al, 2017).

For electrophysiological assays, currents were detected in oocytes using a continuous perfusion system as described previously (Xiao et al, 2021). Briefly, the oocytes were immobilized in basal solution containing 0.15 mM CaCl₂, 3 mM 2-(N-morpholino)-ethanesulfonic acid (MES), and 230 mmol/kg mannitol (pH 7.5 adjusted with Bis-Tris propane) for 5–10 min, until membrane potential became stable. After stability was reached, the currents were recorded in basal solution supplemented with 10 mM HCl or 10 mM HNO₃ (pH 5.5 or 7.5 adjusted with Bis-Tris propane). The oocytes were again perfused with basal solution until membrane potential returned to the initial current level. The Cl⁻ or NO₃⁻ channel-elicited current, i.e., the actual current change caused by Cl⁻ or NO₃⁻ in a single oocyte, was determined as the current recorded in the Cl⁻ or NO₃⁻ solution minus the current recorded in the basal solution. The currents of whole oocytes were recorded using a two-electrode voltage clamp system comprising an AxoClamp 900 A amplifier and a Digidata 1440 A A/D converter (Molecular Devices, Sunnyvale, CA, USA). The electrodes were filled with 3 M KCl.

We conducted Cl⁻ uptake experiments in the oocytes as previously described (Xiao et al, 2021), with modifications. After 48 h of cRNA injection, the oocytes were washed three to five times and then transferred into a culture solution containing Cl⁻ (NaCl) at pH 5.5 or 7.5 for 12 h. Subsequently, the oocytes were harvested, washed three to five times with ice-cold basal solution, and dried at 65 °C for 3 days. Finally, the oocytes were extracted with 1 mL of

ultra-pure water and filtered. Cl⁻ content in the oocytes was detected using ion chromatography (ICS 1100, Thermo Fisher Scientific).

The $NO_3^-$ uptake of GmNPF7.5 in the oocytes was detected using $Na^{15}NO_3$ as previously described (Xiao et al, 2021). At 48 h post-injection, the oocytes were transferred to basal solution containing 10 mM or 0.25 mM $Na^{15}NO_3$. After 12 h of treatment, the oocytes were washed three to five times and dried at 65 °C for 3 days. Then the samples were measured and analyzed using an isotope ratio mass spectrometer (Delta V Advantage, Thermo Fisher Scientific).

For the $Cl^-/NO_3^-$ competition assay, the oocytes were treated in basal solution containing 10 mM NaCl alone or 10 mM NaCl + 10 mM $NaNO_3$ for 12 h. The oocytes were collected, washed three to five times with ice-cold basal solution, and dried. Then, Cl⁻ content was measured in oocytes injected with GmNPF7.5 or water (Xiao et al, 2021). For the $^{15}NO_3^-/Cl^-$ competition assay, the oocytes were treated in basal solution with 10 mM $^{15}NaNO_3$ alone or 10 mM $^{15}NaNO_3$ + 10 mM NaCl for 12 h. Then, $^{15}N$ content was measured in oocytes injected with GmNPF7.5 or water (Xiao et al, 2021).

Xenopus keeping and experiments were approved and supervized by Experimental animal Welfare and Ethics Committee in Nanjing Agricultural University.

## Subcellular localization of GmNPF7.5 protein in *N. benthamiana* leaves

The CDS sequence of *GmNPF7.5* was fused to the GFP fragment and the fused DNA was assembled into pCM1307 vector. Then, pCM1307–GmNPF7.5–GFP and AtCBL1n–mCherry were cotransformed into *N. benthamiana* leaves and fluorescence was detected after 2–3 days of incubation. Then, pCM1307–GFP and AtCBL1n–mCherry were cotransformed as a negative control (Shen et al, 2015).

## GUS activity analysis

An approximately 1.7 kb region of ATG upstream of *GmNPF7.5* was cloned from soybean "Williams 82" DNA using specific primers, and further ligated into pCAMBIA1301 vector, replacing the *35S* promoter and LacZ regions. Then, *GmNPF7.5* promoter–GUS was introduced into *A. rhizogenes* strain "K599" and transformed into soybean hairy roots. For GUS staining, soybean hairy root samples were soaked in staining solution at 37 °C for 2 h (Jefferson et al, 1987), fixed in 3% agarose (m/v), and cut using a vibrating blade microtome (VT1200 S, Leica Biosystems, Nuβloch, Germany). Meanwhile, 1-week-old seedlings were treated with 150 mM NaCl for 6 h, and then stained and cut.

## In vitro kinase assays

The central linker (251–349 amino acids [aa]) of *GmNPF7.5* and the full-length CDS of *PI4Kγ4* were amplified, recombined into the vector pGEX4T-1, and tagged with GST. The recombinant constructs were inducted and purified after transformation into *Escherichia coli* strain "BL21."

In vitro kinase assays were conducted as described previously (Tang et al, 2016). Purified proteins were mixed with 1 μCi (γ-³²P)

ATP and kinase reaction buffer containing 25 mM Tris-HCl (pH 7.5), 5 mM $MgCl_2$, and 0.2 mM ethylenediaminetetraacetic acid at 30 °C for 1 h. Then the reaction was terminated by adding 5× sodium dodecyl sulfate (SDS) loading buffer at 95 °C for 10 min. The incubated proteins were separated by 10% SDS–polyacrylamide gel electrophoresis, and the phosphorylated signals were detected using an imager.

## In vitro pull-down assays

To characterize the interaction of GmNPF7.5 and PI4Kγ4, the CDS encoding the central linker (251–349 amino acids [aa]) of *GmNPF7.5* recombined with GST-tag and full-length CDS of *PI4Kγ4* recombined with His-tag were amplified into the pGEX4T-1 vector and pET28a vector, respectively. The fused plasmids were transformed into *E. coli* strain "BL21" and then induced and purified. The pull-down assays were performed as previously described (Li et al, 2024).

## Yeast two-hybrid assays

Yeast two-hybrid assays were performed as described previously (Wang et al, 2023). The CDS encoding the central linker (251–349 amino acids [aa]) of *GmNPF7.5* was cloned into the pGBKT7–BD vector, and the full-length CDS of *PI4Kγ4* was amplified into the pGADT7–AD vector. The recombinational plasmids were transformed into yeast strain 'AH109.' The resulting transformants were selected on SD–Leu–Trp medium at 30 °C for 2–4 days. The positive clones were cultured on SD–Leu–Trp and SD–Leu–Trp–His–Ade medium at 30 °C for 3–5 days.

## BiFC assays

BiFC assays were performed as described previously (Wang et al, 2021b). For yellow fluorescent protein (YFP) analysis, the full-length CDSs of *GmNPF7.5* and *PI4Kγ4* were cloned into cYFP and nYFP, respectively. Then the fused plasmids were transformed into *N. benthamiana* leaves and the YFP fluorescence signal was detected after 2–3 days of incubation.

## FRET–FLIM assays

FRET–FLIM assays were performed as previously described (Li et al, 2024). Briefly, the full-length CDSs of *GmNPF7.5* and *PI4Kγ4* were inserted into pCM1307–GFP vector and pCM1307–mCherry vector, respectively. Then, either GmNPF7.5–GFP or GmNPF7.5–GFP and PI4Kγ4-mCherry were transfected into *N. benthamiana* leaves. The leaves were subjected to fluorescence confocal imaging and fluorescence lifetime imaging using a TCS-SP8 confocal laser scanning microscope (Leica Biosystems).

## Co-IP assays

The CDS encoding the central linker (251–349 amino acids [aa]) of *GmNPF7.5* recombined with GFP-tag, and the full-length CDS of *PI4Kγ4* recombined with flag-tag were amplified into the pCM1307 vector. The fused plasmids were transformed into *GV3101* and cotransformed into *N. benthamiana* leaves. Then, total protein was extracted from *N. benthamiana* leaves and Co-IP assays were performed as previously described (Wang et al, 2021b).

## Statistical analyses

Significant differences between two groups were analyzed using Student's $t$ tests, and those among multiple samples were evaluated using one-way or two-way analysis of variance, followed by Tukey's test.

# Data availability

All relevant data will be made available from the authors upon request. The raw sequencing data for the 198 soybean accessions reported elsewhere (Zhang et al, 2021). The RNA-seq data are deposited into the Genome Sequence Archive (GSA) database in the National Genomics Data Center under accession number SAMC4041293 − SAMC4041298 of PRJCA028492.

The source data of this paper are collected in the following database record: biostudies:S-SCDT-10_1038-S44318-024-00357-1.

# Peer review information

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

## Acknowledgements

The authors thank Xuelu Wang (Henan University) for providing pUB–GFP vector and pGmUbi:3×flag vector. The authors also thank Bincheng Sun (Hulunbuir Institute of Agricultural and Animal Husbandry Science) for kindly providing the "Mengdou 1137" and "Dengke 5" soybean cultivars, Xinlei Liu (Heilongjiang Academy of Agricultural Sciences) for kindly providing the "Heinong 84" cultivar, and Jinxing Wang (Heilongjiang Academy of Agricultural Sciences) for kindly providing the "Suinong 52" cultivar. This research was supported by grants from the National Key Research & Development Program of China (No. 2022YFD1201700 and No. 2022YFA1303400), Fundamental Research Funds for the Central Universities (No. KJJQ2024007), and the National Natural Science Foundation of China (No. 32270268 to LS and No. 32270301 to QZ).

## Author contributions

**Yunzhen Wu**: Data curation; Formal analysis; Investigation; Writing—original draft; Writing—review and editing. **Jingya Yuan**: Data curation; Formal analysis; Investigation; Writing—original draft; Writing—review and editing. **Like Shen**: Conceptualization; Supervision; Writing—original draft; Writing—review and editing. **Qinxue Li**: Resources. **Zhuomeng Li**: Investigation. **Hongwei Cao**: Formal analysis; Investigation. **Lin Zhu**: Formal analysis; Investigation. **Dan Liu**: Formal analysis; Investigation. **Yalu Sun**: Formal analysis; Investigation. **Qianru Jia**: Resources. **Huatao Chen**: Resources. **Wubin Wang**: Resources. **Jörg Kudla**: Writing—review and editing; Data discussion. **Wenhua Zhang**: Supervision; Writing—original draft; Writing—review and editing. **Junyi Gai**: Conceptualization; Supervision. **Qun Zhang**: Conceptualization; Supervision; Writing—original draft; Writing—review and editing.

Source data underlying figure panels in this paper may have individual authorship assigned. Where available, figure panel/source data authorship is listed in the following database record: biostudies:S-SCDT-10_1038-S44318-024-00357-1.

## Disclosure and competing interests statement

The authors declare no competing interests.

# Expanded View Figures

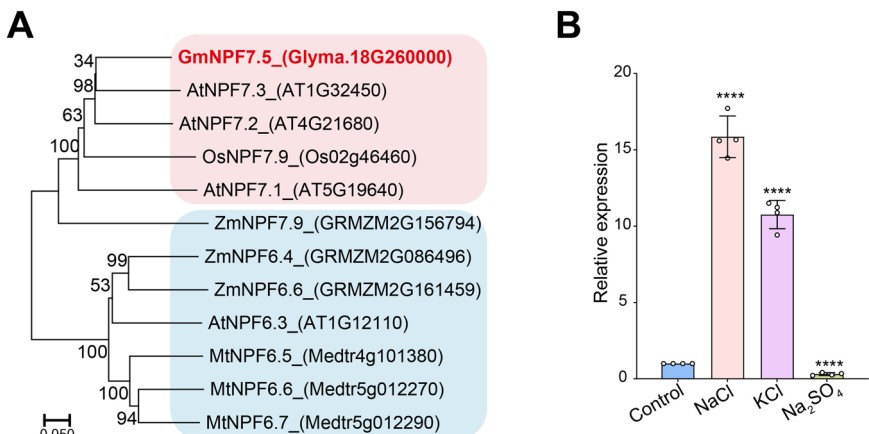

**Figure EV1.   Phylogenetic tree and transcript levels of *GmNPF7.5* in soybean plants under indicated conditions.**

(A) Phylogenic analysis of NPF6 and NPF7 from *Arabidopsis*, rice, maize, and soybean. The phylogenetic tree was constructed using MEGA7. (B) Transcript levels of *GmNPF7.5* in soybean plants under the indicated conditions. *GmELF* was used as an internal control. Data are means ± standard error (SEM). Significance was determined using a two-sided Student's $t$ test (****$P < 0.0001$), $n = 4$ (4 biological replicates). $P$ values $< 0.0001$ (NaCl vs Control), $< 0.0001$ (KCl vs Control), $< 0.0001$ (Na$_2$SO$_4$ vs Control). Source data are available online for this figure.

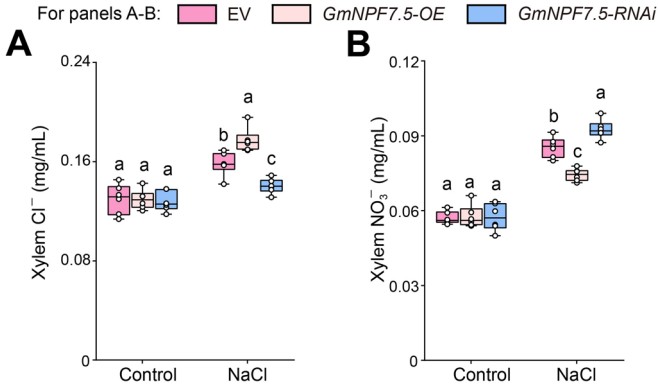

**Figure EV2.** **Comparison of Cl⁻ and NO₃⁻ content levels in xylem sap of transgenic soybean lines with hairy roots harboring the indicated constructs.**

(A, B) Cl⁻ concentration (A), NO₃⁻ concentration (B) in xylem sap of $GmNPF7.5$ overexpression ($GmNPF7.5$-OE) or knockdown ($GmNPF7.5$-RNAi) hairy root transgenic soybean lines. Data in (A, B) are means ± SEM ($n = 6$ independent biological replicates). Significance was determined using one-way ANOVA, followed by Tukey's test. Different letters indicate significant differences ($P < 0.05$). (A) Control: $P$ values = 0.9992 (EV vs $GmNPF7.5$-OE), 0.9411 (EV vs $GmNPF7.5$-RNAi), 0.9535 ($GmNPF7.5$-OE vs $GmNPF7.5$-RNAi). (The following is the same order). NaCl: $P$ values = 0.0034, 0.0051, <0.0001. (B) Control: $P$ values = 0.9729, 0.9886, 0.9965. NaCl: $P$ values = 0.0001, 0.0116, <0.0001. For box plots: the whiskers represent maximum and minimum values, and boxes represent he upper quartile, median, and lower quartile, dots represent data points. Source data are available online for this figure.

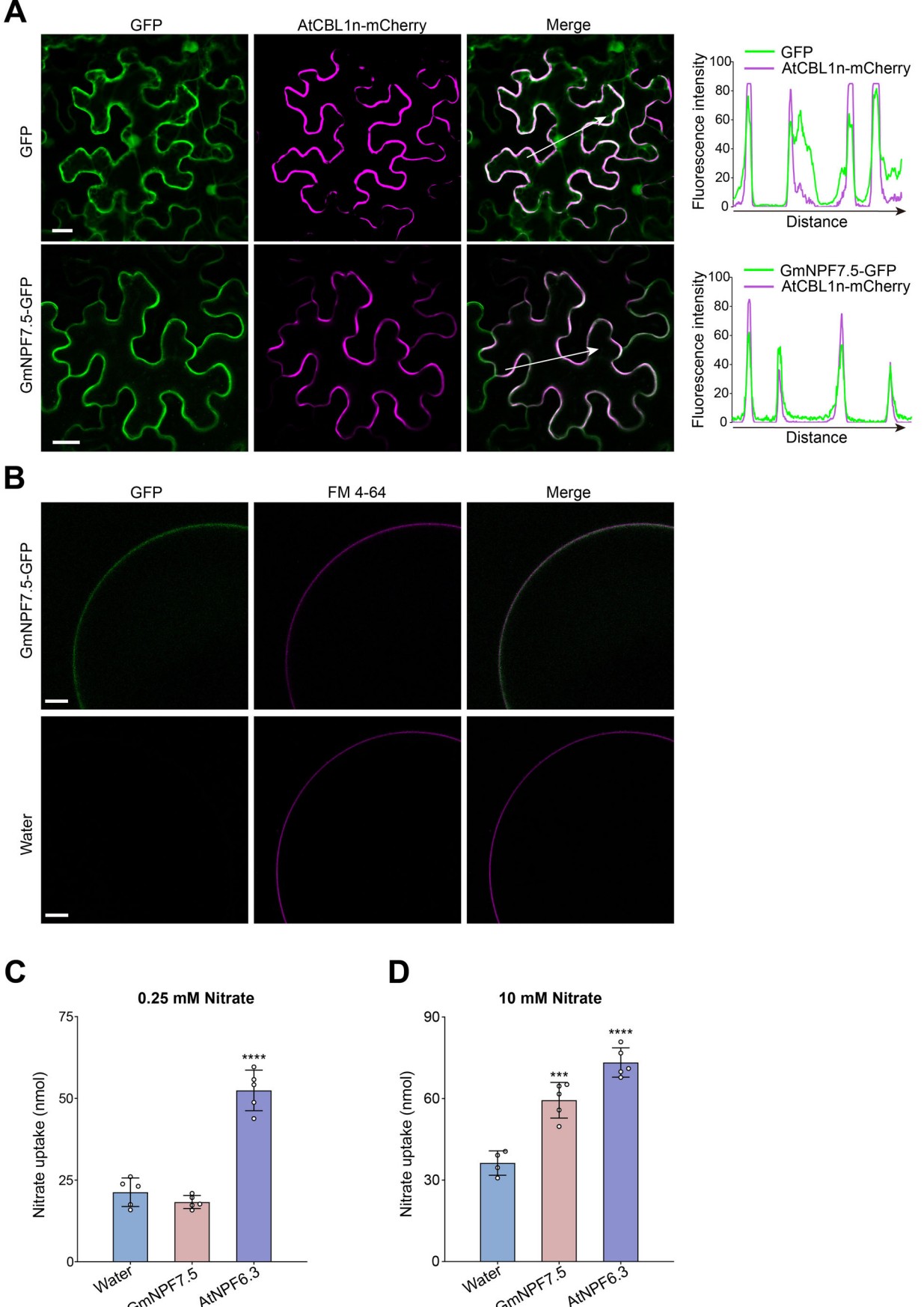

◀ **Figure EV3. Localization of GmNPF7.5 and NO$_3^-$ uptake assay results.**

(A) Subcellular localization of GmNPF7.5–GFP in *N. benthamiana* leaves, shown by confocal images (left) and fluorescence intensity (right, arrows). Scale bars, 20 μm. (B) GmNPF7.5–GFP was localized to the plasma membrane in *Xenopus* oocytes. Scale bars, 100 μm. (C) High-affinity (0.25 mM) and (D) low-affinity (10 mM) NO$_3^-$ uptake assay using oocytes expressing GmNPF7.5 and AtNPF6.3 (positive control). Data are means ± standard error (SEM). Significance was determined using a two-sided Student's *t* test (***$P < 0.001$, ****$P < 0.0001$; $n = 4$–5, each replicate contained 2 oocytes). (C), *P* values = 0.1993 (GmNPF7.5 vs Water), < 0.0001 (AtNPF6.3 vs Water). (D) *P* values = 0.0006 (GmNPF7.5 vs Water), <0.0001 (AtNPF6.3 vs Water). Source data are available online for this figure.

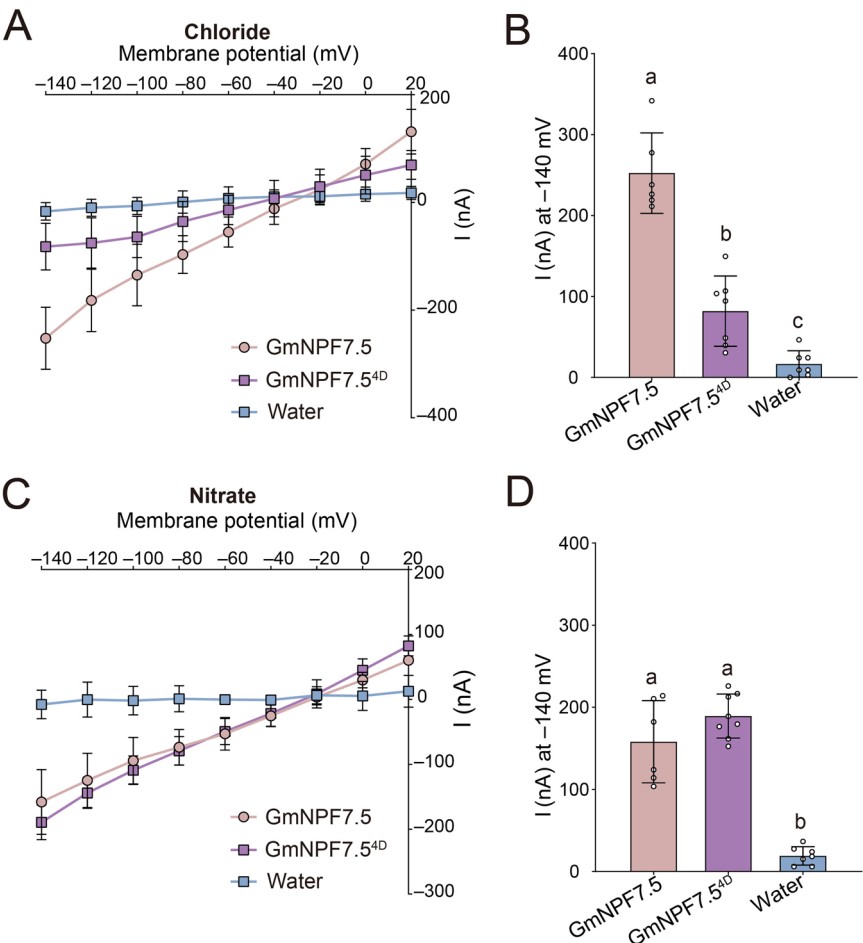

**Figure EV4. Effects of phosphomimetic GmNPF7.5 in oocytes.**

(A) I–V relationship for *Xenopus* oocytes expressing GmNPF7.5, GmNPF7.5[4D], and water in basal solution containing 10 mM Cl⁻ at pH 5.5. (B) Cl⁻-elicited currents at –140 mV recorded from *Xenopus* oocytes in (A). (C) I–V relationship for *Xenopus* oocytes expressing GmNPF7.5, GmNPF7.5[4D], and water in basal solution containing 10 mM $NO_3^-$ at pH 5.5. (D) $NO_3^-$-elicited currents at –140 mV recorded from *Xenopus* oocytes in (C). Data in (B) and (D) are means ± SEM ($n$ = 6–8 single oocytes). Significance was determined using one-way ANOVA, followed by Tukey's test. Different letters indicate significant differences ($P < 0.05$). (C), $P$ values < 0.0001 (GmNPF7.5 vs GmNPF7.5[4D]), <0.0001 (GmNPF7.5 vs Water), 0.0147 (GmNPF7.5[4D] vs Water). (D), $P$ values = 0.193 (GmNPF7.5 vs GmNPF7.5[4D]), <0.0001 (GmNPF7.5 vs Water), <0.0001 (GmNPF7.5[4D] vs Water). Source data are available online for this figure.

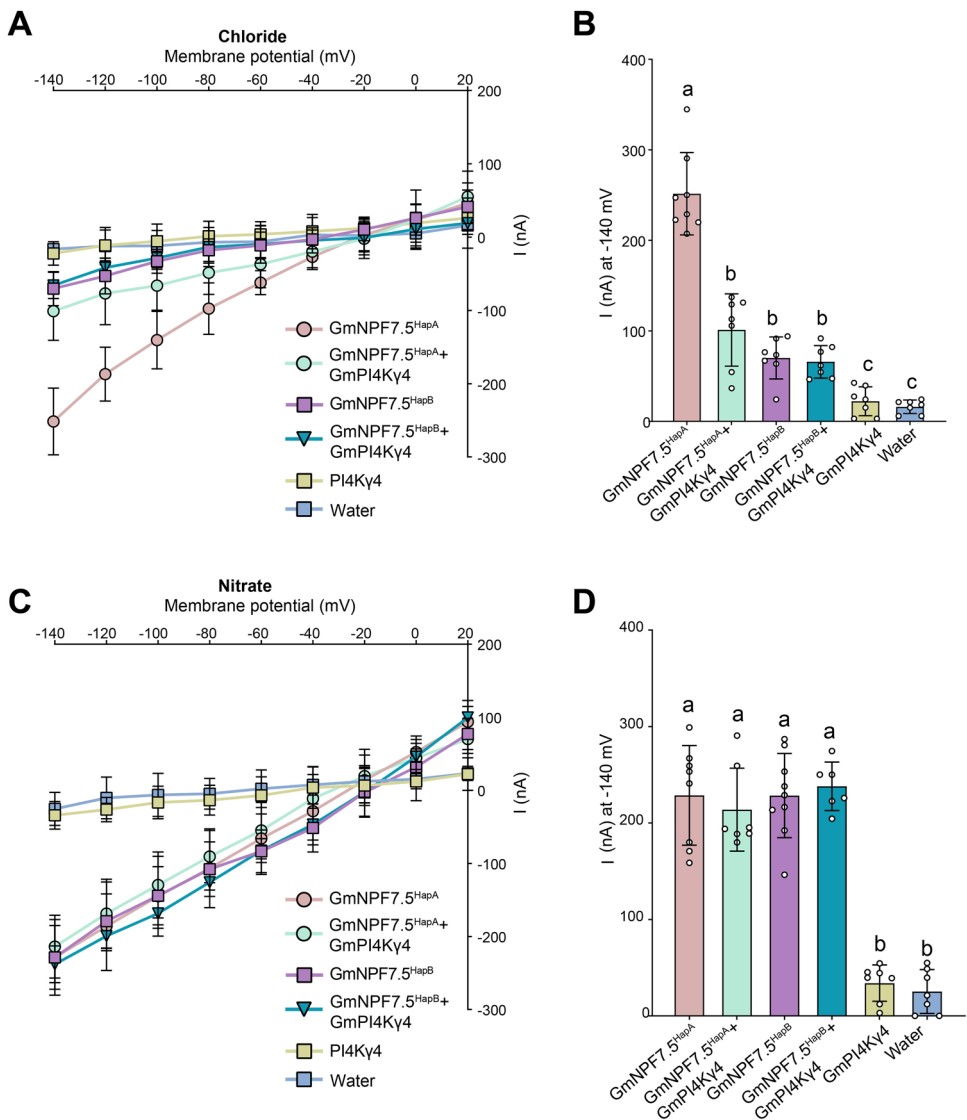

**Figure EV5. Effects of GmPI4Kγ4 on the transport activity of two GmNPF7.5 haplotypes in oocytes.**

(A, C) I–V relationship for *Xenopus* oocytes expressing GmNPF7.5$^{HapA}$, GmNPF7.5$^{HapB}$, GmPI4Kγ4, GmNPF7.5$^{HapA}$ + GmPI4Kγ4 or GmNPF7.5$^{HapB}$ + GmPI4Kγ4 in basal solution with addition of 10 mM Cl$^-$ (A) or 10 mM NO$_3^-$ (C) at pH 5.5 ($n = 6$–9). (B, D) Cl$^-$-elicited or NO$_3^-$-elicited currents at –140 mV recorded from *Xenopus* oocytes in (A) or (C). Data in (B) and (D) are means ± SEM ($n = 6$–9 single oocytes). Significance was determined using one-way ANOVA, followed by Tukey's test. Different letters indicate significant differences ($P < 0.05$). (B) $P$ values < 0.0001 (GmNPF7.5$^{HapA}$ vs GmNPF7.5$^{HapA}$ + GmPI4Kγ4), <0.0001 (GmNPF7.5$^{HapA}$ vs GmNPF7.5$^{HapB}$), <0.0001 (GmNPF7.5$^{HapA}$ vs GmNPF7.5$^{HapB}$ + GmPI4Kγ4), <0.0001 (GmNPF7.5$^{HapA}$ vs GmPI4Kγ4), < 0.0001 (GmNPF7.5$^{HapA}$ vs Water), 0.363 (GmNPF7.5$^{HapA}$ + GmPI4Kγ4 vs GmNPF7.5$^{HapB}$), 0.2314 (GmNPF7.5$^{HapA}$ + GmPI4Kγ4 vs GmNPF7.5$^{HapB}$ + GmPI4Kγ4), 0.0001 (GmNPF7.5$^{HapA}$ + GmPI4Kγ4 vs GmPI4Kγ4), <0.0001 (GmNPF7.5$^{HapA}$ + GmPI4Kγ4 vs Water), 0.9998 (GmNPF7.5$^{HapB}$ vs GmNPF7.5$^{HapB}$ + GmPI4Kγ4), 0.0405 (GmNPF7.5$^{HapB}$ vs GmPI4Kγ4), 0.015 (GmNPF7.5$^{HapB}$ vs Water), 0.0767 (GmNPF7.5$^{HapB}$ + GmPI4Kγ4 vs GmPI4Kγ4), 0.0302 (GmNPF7.5$^{HapB}$ + GmPI4Kγ4 vs Water), 0.9987 (GmPI4Kγ4 vs Water). (The following is the same order). (D), $P$ values = 0.9721, >0.9999, 0.9972, <0.0001, <0.0001, 0.9701, 0.854, <0.0001, <0.0001, 0.9966, <0.0001, <0.0001, <0.0001, 0.9978. Source data are available online for this figure.

