## [Peer Review File · The EMBO Journal]

A phosphorylation-regulated NPF transporter determines salt tolerance by mediating chloride uptake in soybean plants

Yunzhen Wu, Jingya Yuan, Like Shen, Qinxue Li, Zhuomeng Li, Hongwei Cao, Lin Zhu, Dan Liu, Yalu Sun, Qianru Jia, Huatao Chen, Wubin Wang, Jörg Kudla, Wenhua Zhang, Junyi Gai, and Qun Zhang

Corresponding author(s): Qun Zhang (zhangqun@njau.edu.cn) , Like Shen (likeshen@njau.edu.cn), Junyi Gai (sri@njau.edu.cn)

Review Timeline:

Submission Date:	14th Aug 24
Editorial Decision:	28th Sep 24
Revision Received:	7th Nov 24
Editorial Decision:	29th Nov 24
Revision Received:	9th Dec 24
Accepted:	13th Dec 24

Editor: William Teale

Transaction Report:

Dear Prof. Zhang,

Thank you again for the submission of your manuscript entitled "A phosphorylation-regulated NPF transporter determines salt tolerance by mediating chloride uptake in soybean plants" and for your patience during the review process. We have now received the reports from the referees, which I copy below.

As you can see from their comments, referee #1 requests some further experimental work and textual clarification of some key points. That said, both referees point out the timeliness and solidity of your work.

Based on the overall interest expressed in the reports, I would therefore like to invite you to address the comments of all referees in a revised version of the manuscript. I should add that it is The EMBO Journal policy to allow only a single major round of revision and that it is therefore important to resolve the main concerns at this stage. I believe the concerns of the referees are reasonable and addressable, but please contact me if you have any questions, need further input on the referee comments or if you anticipate any problems in addressing any of their points. Please, follow the instructions below when preparing your manuscript for resubmission.

I would also like to point out that as a matter of policy, competing manuscripts published during this period will not be taken into consideration in our assessment of the novelty presented by your study ("scooping" protection). We have extended this 'scooping protection policy' beyond the usual 3 month revision timeline to cover the period required for a full revision to address the essential experimental issues. Please contact me if you see a paper with related content published elsewhere to discuss the appropriate course of action.

Again, please contact me at any time during revision if you need any help or have further questions.

Thank you very much again for the opportunity to consider your work for publication. I look forward to your revision.

Best regards,

William

William Teale, Ph.D.
Editor
The EMBO Journal

When submitting your revised manuscript, please carefully review the instructions below and include the following items:

- 1) a .docx formatted version of the manuscript text (including legends for main figures, EV figures and tables). Please make sure that the changes are highlighted to be clearly visible.
- 2) individual production quality figure files as .eps, .tif, .jpg (one file per figure).
- 3) a .docx formatted letter INCLUDING the reviewers' reports and your detailed point-by-point response to their comments. As part of the EMBO Press transparent editorial process, the point-by-point response is part of the Review Process File (RPF), which will be published alongside your paper.
- 4) a complete author checklist, which you can download from our author guidelines ([https://wol-prod-cdn.literatumonline.com/pb-assets/embo-site/Author Checklist%20-%20EMBO%20J-1561436015657.xlsx](https://wol-prod-cdn.literatumonline.com/pb-assets/embo-site/Author%20Checklist%20-%20EMBO%20J-1561436015657.xlsx)). Please insert information in the checklist that is also reflected in the manuscript. The completed author checklist will also be part of the RPF.
- 5) Please note that all corresponding authors are required to supply an ORCID ID for their name upon submission of a revised manuscript.
- 6) We require a 'Data Availability' section after the Materials and Methods. Before submitting your revision, primary datasets produced in this study need to be deposited in an appropriate public database, and the accession numbers and database listed under 'Data Availability'. Please remember to provide a reviewer password if the datasets are not yet public (see <https://www.embopress.org/page/journal/14602075/authorguide#datadeposition>). If no data deposition in external databases is

needed for this paper, please then state in this section: This study includes no data deposited in external repositories. Note that the Data Availability Section is restricted to new primary data that are part of this study.

Note - All links should resolve to a page where the data can be accessed.

8) For data quantification: please specify the name of the statistical test used to generate error bars and P values, the number (n) of independent experiments (specify technical or biological replicates) underlying each data point and the test used to calculate p-values in each figure legend. The figure legends should contain a basic description of n, P and the test applied. Graphs must include a description of the bars and the error bars (s.d., s.e.m.).

9) We would also encourage you to include the source data for figure panels that show essential data. Numerical data can be provided as individual .xls or .csv files (including a tab describing the data). For 'blots' or microscopy, uncropped images should be submitted (using a zip archive or a single pdf per main figure if multiple images need to be supplied for one panel). Additional information on source data and instruction on how to label the files are available at .

10) We replaced Supplementary Information with Expanded View (EV) Figures and Tables that are collapsible/expandable online (see examples in <https://www.embopress.org/doi/10.15252/embj.201695874>). A maximum of 5 EV Figures can be typeset. EV Figures should be cited as 'Figure EV1, Figure EV2" etc. in the text and their respective legends should be included in the main text after the legends of regular figures.

12) Our journal encourages inclusion of *data citations in the reference list* to directly cite datasets that were re-used and obtained from public databases. Data citations in the article text are distinct from normal bibliographical citations and should directly link to the database records from which the data can be accessed. In the main text, data citations are formatted as follows: "Data ref: Smith et al, 2001" or "Data ref: NCBI Sequence Read Archive PRJNA342805, 2017". In the Reference list, data citations must be labeled with "[DATASET]". A data reference must provide the database name, accession number/identifiers and a resolvable link to the landing page from which the data can be accessed at the end of the reference. Further instructions are available at .

13) In order to increase the reproducibility and reach of your work, The EMBO Journal includes a table of reagents that were used in the study. Please provide this along with your revisions.

Further instructions for preparing your revised manuscript:

We realize that it is difficult to revise to a specific deadline. In the interest of protecting the conceptual advance provided by the work, we recommend a revision within 3 months (27th Dec 2024). Please discuss the revision progress ahead of this time with the editor if you require more time to complete the revisions. Use the link below to submit your revision:

Referee #1:

This work shows that the soybean protein NPF7.5 is a dual transporter mediating chloride and nitrate uptake into roots, similarly to related proteins in other plant species. However, an NPF7.5 haplotype was discovered in which chloride transport was largely suppressed. Further, the kinase PI4KY4 phosphorylated NPF7.5 to inhibit chloride transport without affecting nitrate uptake. The differential substrate selectivity of NPF7.5 haplotypes could be functionally linked to the amount of accumulated chloride under salinity (NaCl) stress and to differential halotolerance.

The broad selectivity of NPF/NTR proteins and the antagonistic effect of chloride in nitrate uptake have been known for some time. However, the other findings reported in this work are novel and add new layers of biological complexity (natural variation) and of protein regulation (substrate specificity modulated by protein phosphorylation). The findings are also of potential interest to plant biotechnologist as a tool to reduce Cl/NO₃ antagonism in plant nutrition.

The experimental setups are correct and the conclusions are sound. One experiment I think is missing is testing the effect of the kinase PI4KY4 on the NPF7.5-HapB haplotype, to further confirm that the specific NO₃ transport of this protein is not altered by the kinase.

Other than that, I have only a few questions and suggestions aiming to improve the formal aspects of the manuscript and the depth of the Discussion.

1. If I understand the GWS right, the HapA (competent for chloride transport) is predominant in the elite soybean lines. What was the prevalence of the HapB in the 198 accessions assayed? Is there any correlation between a given haplotype and being more or less domesticated? Has the ability of NPF7.5 to take up chloride been acquired during domestication?
2. Revise the Abstract to make it clear whether the dominant allele of GmNPF7.5 is the haplotype leading to greater or lower Cl uptake. Also, in most experiments authors only state GmNPF7.5 gene or protein without specifying the protein variant used (HapA/B) and the haplotype of the recipient plant.
3. Was the expression of NPF7.5 and PI4KY4 similarly enhanced by NaCl in the two plant haplotypes?
4. What was the criteria for choosing the promoter region for construct ProGmNPF7.5:GUS?

5. Fig 1e, explain differences between left and right panels in each condition. Are they different parts of the same root? What developmental zones are shown, including the cross-section?
6. Section 'GmNPF7.5 negatively regulates salt tolerance', lines 130-154. What haplotype was over-expressed or suppressed?
7. Ln 147-148. I am not convinced that Cl/NO₃ ratios could be used similarly to the Na/K ratio often used as an indicator of sodicity stress because contrary to Na, K and Cl, nitrate is metabolized into organic matter and thus the absolute NO₃ content may vary greatly depending on the relative rates of uptake and usage.
8. Fig 3, electrophysiology and Cl/NO₃ uptake. I do not fully understand the experimental conditions used for TEVC. The basal solution contained 3 mM MES as the pH buffer. Then, basal solutions supplemented with 10 mM HCl or HNO₃ at pH 5.5 or 7.5 were perfused. How were these solutions prepared at different pH with the same amount of MES? What salt was used for the Cl uptake experiments (ln 464-470)? Again, what haplotype was used for (only specified in panels i,j)?
9. Ln 196. State that the Y2H screening was done with only the central linker domain of GmNPF7.5. The central linker (CL) domain of GmNPF7.5 needs to be defined when first mentioned (Ln 202-203).
10. Ln 214-221. Over-expression of PI4KY4 in soybean. In what plant haplotype?
11. Ln 509. Kinase assay buffer with 1 M MgCl₂?

Referee #2:

In this work, Wu et al conducted a GWAS and transcriptomic analysis to identify the dominant gene locus influencing Cl⁻ homeostasis in soybean plants and investigated mechanistic basis of regulation of Cl⁻ uptake by plant roots. They demonstrated that salt stress induced expression of GmNPF7.5, an anion transporter with a dual affinity for chloride and nitrate. At the same time, salt stress also induced expression of GmPI4Ky4 that phosphorylated GmNPF7.5 and inhibited its Cl⁻ uptake without affection NO₃⁻ uptake. Thus, GmPI4Ky4 represents some sort of a metabolic "switch" that modulates selectivity of the anion channel and improve N/Cl ratio in plants, while reducing overall Cl accumulation.

Overall, the paper is well written and performed at highest possible level. All experiments are well thought of and executed, and all conclusions are supported by the actual data. The above regulatory mechanisms reported in this work is highly novel and may be a game-changer for breeding salt-tolerant cultivars. It is my pleasure therefore to recommend this work being published essentially as it is.

Point-by-point responses to Reviewers

Referee #1: (Comments for the Author):

This work shows that the soybean protein NPF7.5 is a dual transporter mediating chloride and nitrate uptake into roots, similarly to related proteins in other plant species. However, an NPF7.5 haplotype was discovered in which chloride transport was largely suppressed. Further, the kinase PI4KY4 phosphorylated NPF7.5 to inhibit chloride transport without affecting nitrate uptake. The differential substrate selectivity of NPF7.5 haplotypes could be functionally linked to the amount of accumulated chloride under salinity (NaCl) stress and to differential halotolerance.

The broad selectivity of NPF/NTR proteins and the antagonistic effect of chloride in nitrate uptake have been known for some time. However, the other findings reported in this work are novel and add new layers of biological complexity (natural variation) and of protein regulation (substrate specificity modulated by protein phosphorylation). The findings are also of potential interest to plant biotechnologists as a tool to reduce Cl/NO₃ antagonism in plant nutrition.

The experimental setups are correct and the conclusions are sound. One experiment I think is missing is testing the effect of the kinase PI4KY4 on the NPF7.5-HapB haplotype, to further confirm that the specific NO₃ transport of this protein is not altered by the kinase.

Response: Thank you for the comments. The additional electrophysiological assay was performed to detect the effect of GmPI4K γ 4 on the GmNPF7.5^{HapB} haplotype. Co-expressing GmPI4K γ 4 with GmNPF7.5^{Hap} in oocytes had no effect on the GmNPF7.5^{HapB}-mediated NO₃⁻ transport. We described the result in the revised manuscript (Lines 251–253, Page 7–8) and set the diagrams as Fig. EV5.

Other than that, I have only a few questions and suggestions aiming to improve the formal aspects of the manuscript and the depth of the Discussion.

1. If I understand the GWS right, the HapA (competent for chloride transport) is predominant

in the elite soybean lines. What was the prevalence of the HapB in the 198 accessions assayed? Is there any correlation between a given haplotype and being more or less domesticated? Has the ability of NPF7.5 to take up chloride been acquired during domestication?

Response: The prevalence of the HapB in the 198 accessions assayed in GWAS is approximately 68%. To determine whether there is any correlation between the haplotype variation and domestication, we calculated the proportion of two haplotypes in wild species (WS), landraces (LS) and cultivars (CS), respectively (Figure 1 for reviewers). There was no significant difference in proportion of the two haplotypes among wild species (WS), landraces (LS) and cultivars (CS). Considering that the Cl⁻ transport activity of GmNPF7.5 was determined by the SNP variation, we proposed that the haplotype variation of GmNPF7.5 and its Cl⁻ uptake ability may be not relevant to domestication.

Figure 1 for reviewers. The proportion of two haplotypes in wild species (WS), landraces (LS) and cultivars (CS) populations. In this analysis, there are 36 accessions from WS, 38 accessions from LS and 198 accessions from CS.

2. *Revise the Abstract to make it clear whether the dominant allele of GmNPF7.5 is the*

haplotype leading to greater or lower Cl uptake. Also, in most experiments authors only state GmNPF7.5 gene or protein without specifying the protein variant used (HapA/B) and the haplotype of the recipient plant.

Response: According to the reviewer's request, we revised the abstract (Line41–47, Page 2) and specified the haplotypes of *GmNPF7.5* and the recipient plants in necessary places throughout the revised manuscript. To state the haplotypes clearly, we added the descriptions in Methods section: “Unless specifically stated, the recipient plants were Williams 82 (a cultivar with HapA).” (Lines 422–423, Page 12), and “If not specifically indicated, the *GmNPF7.5* represents *GmNPF7.5*^{HapA} in most experiments.” (Lines 401–402, Page 12).

3. Was the expression of NPF7.5 and PI4Kγ4 similarly enhanced by NaCl in the two plant haplotypes?

Response: To address this concern, we randomly chose seven cultivars from each of the two haplotypes. Reverse transcription quantitative PCR (RT-qPCR) was performed to detect the NaCl-induced transcriptional changes of *GmNPF7.5* and *GmPI4Kγ4* in each cultivar. As shown in following diagrams, the two haplotypes showed similar transcriptional changes of *GmNPF7.5* in response to NaCl stress. This may be consistent with our GWAS result, because we didn't identify any SNP variation associated with Cl⁻ content in *GmNPF7.5* promoter region. In addition, the expression of *GmPI4Kγ4* in HapA increased more than that in HapB under NaCl treatment (Figure 2 for reviewers). Due to the limited paper length, these results were not included in the revised manuscript.

Figure 2 for reviewers. The expression of *GmNPF7.5* (A) and *GmPI4Ky4* (B) in the two haplotypes under NaCl treatment. 14-d-old seedlings were exposed to 150 mM NaCl for 6 h. The gene transcriptional changes represented the ratio of transcriptional level after NaCl treatment to the corresponding level before treatment. *GmELF* was used as an internal control. Statistical significance was determined using Student's *t*-test (n = 7).

4. What was the criteria for choosing the promoter region for construct *ProGmNPF7.5:GUS*?

Response: Usually, we chose 2~2.5 kb region upstream ATG as the promoter of a gene. In this study, we fail to clone the intact 2.0 kb region upstream ATG of *GmNPF7.5*, possibly due to a 45-bp highly repeated sequence (ATA) located at 1700–1745 bp upstream ATG. So we chose the 1.7-kb region upstream ATG as the promoter of *GmNPF7.5* to construct *ProGmNPF7.5:GUS*. Considering that the GUS signal was increased by NaCl stress, which was consistent with the transcriptional analysis, we speculated that the 1.7-kb region can act well as the promoter of *GmNPF7.5*.

5. Fig 1e, explain differences between left and right panels in each condition. Are they different parts of the same root? What developmental zones are shown, including the cross-section?

Response: In each condition, the left panels showed the root tip zones containing root cap,

cell division zone, elongation zone and maturation zone. The right panels showed the middle part of roots, which also belong to maturation zone. The cross-sections were made using the roots of maturation zone. We added these information in the Figure legend (Lines 765–768, Page 22).

6. Section 'GmNPF7.5 negatively regulates salt tolerance', lines 130-154. What haplotype was over-expressed or suppressed?

Response: In this experiment GmNPF7.5^{HapA} was overexpressed or suppressed. We have specified the haplotype (HapA) in Figure 2, Appendix Figure S1 and corresponding parts in main text (Lines 778–780, Page 22).

7. Ln 147-148. I am not convinced that Cl/NO3 ratios could be used similarly to the Na/K ratio often used as an indicator of sodicity stress because contrary to Na, K and Cl, nitrate is metabolized into organic matter and thus the absolute NO3 content may vary greatly depending on the relative rates of uptake and usage.

Response: This is a rational and constructive comment that convinced us. So, we have deleted the Cl⁻/NO₃⁻ ratios in the Figures (Figure EV2, Appendix Figure S1, Appendix Figure S3, Appendix Figure S5 and Appendix Figure S7) and the corresponding description in the revised manuscript (Line 156, Page 5 and Lines 318–320, Page 9).

8. Fig 3, electrophysiology and Cl/NO3 uptake. I do not fully understand the experimental conditions used for TEVC. The basal solution contained 3 mM MES as the pH buffer. Then, basal solutions supplemented with 10 mM HCl or HNO3 at pH 5.5 or 7.5 were perfused. How were these solutions prepared at different pH with the same amount of MES? What salt was used for the Cl uptake experiments (ln 464-470)? Again, what haplotype was used for (only specified in panels i,j)?

Response: Although the amount of MES in different solutions were same, the addition of HCl or HNO₃ introduced H⁺ into the solutions. So, we used Bis-Tris propane to adjust pH. We have added this description in the text (Lines 466–471, Page 13–14).

NaCl was used for the Cl⁻ uptake experiments. We have added this description in the text (Line 480–482, Page 14).

In Figure 3, the GmNPF7.5 in A–H, K and L. are HapA. we have now specified the haplotypes in Figure 3 in the revised manuscript (Lines 790–803, Page 23).

9. Ln 196. State that the Y2H screening was done with only the central linker domain of GmNPF7.5. The central linker (CL) domain of GmNPF7.5 needs to be defined when first mentioned (Ln 202-203).

Response: As the reviewer's suggestion, we stated that only the central linker (CL) domain of GmNPF7.5 was used as the bait in Y2H screening and defined the CL and GmNPF7.5CL abbreviation when first mentioned (Lines 202–203 and Lines 208–209, Page 6).

10. Ln 214-221. Over-expression of PI4KY4 in soybean. In what plant haplotype?

Response: We overexpressed *GmPI4Ky4* in 'Williams 82' (a cultivar with GmNPF7.5^{HapA}). We have added this information in the text and figure legends (Lines 830–832, Page 24 and Lines 859–861, Page 24).

11. Ln 509. Kinase assay buffer with 1 M MgCl₂?

Response: We apologized for this mistake. The concentration of MgCl₂ in stock solution was 1 M. Its working concentration in kinase assay buffer was 5 mM. And We revised the description accordingly in the text (Lines 525–527, Page 15).

Referee #2:

In this work, Wu et al conducted a GWAS and transcriptomic analysis to identify the dominant gene locus influencing Cl⁻ homeostasis in soybean plants and investigated mechanistic basis of regulation of Cl⁻ uptake by plant roots. They demonstrated that salt stress induced expression of GmNPF7.5, an anion transporter with a dual affinity for chloride and nitrate. At the same time, salt stress also induced expression of GmPI4Ky4 that phosphorylated GmNPF7.5 and inhibited its Cl⁻ uptake without affection NO₃⁻ uptake. Thus, GmPI4Ky4 represents some sort of a metabolic "switch" that modulates selectivity of the anion channel and improve N/Cl ratio in plants, while reducing overall Cl accumulation.

Overall, the paper is well written and performed at highest possible level. All experiments are well thought of and executed, and all conclusions are supported by the actual data. The above regulatory mechanisms reported in this work is highly novel and may be a gamechanger for breeding salt-tolerant cultivars. It is my pleasure therefore to recommend this work being published essentially as it is.

Response: Thank you for the comments.

Dear Prof. Zhang,

Thank you for submitting the revised version of your manuscript, which addresses the concerns of the referees. This revised version has now been re-reviewed; I attach the second referee reports to the bottom of this mail. As you will see, you have addressed the referees' concerns to their satisfaction. Before I can finally accept the manuscript, there are some remaining editorial points which need to be addressed. In this regard, would you please:

- add an Orcid for Dr Gai,
- acknowledge funding from National Natural Science Foundation of China - 32270268 in our online submission system,
- rename the Conflict of Interest section the "Disclosure Statement and Competing Interests" statement,
- remove the author credit section from the manuscript file,
- convert the Appendix file to PDF format; Appendix figures should only be compiled in Appendix PDF, there is no need to upload them individually; on title page there should be the subtitle "Appendix for A phosphorylation-regulated NPF transporter determines salt tolerance by mediating chloride uptake in soybean plants" and a table of contents with page numbers for each listed figure and table; Table EV1 (labeled as Supplementary Table 1) should be included in Appendix PDF with the nomenclature Appendix Table S1 throughout Appendix and ms file,
- save Source data files in a scheme of one figure per folder and then upload as .zip files. E.g. all the Source data files for figure 1 need to be saved in a single folder and this needs to be zipped and then uploaded as "SD figure 1.zip" file. For EV and/or appendix figures, ZIP together all source data,
- clarify and provide source data for Appendix Fig S3A as our analysis shows that it may contain empty cells in the 1st and 3rd panels,
- provide specific URLs for SAMC4041293–SAMC4041298 of PRJCA028492 datasets in the data availability statement,
- provide exact p values in the legends of figures 2b-d, f-h; 3e-f, k, l; 4e, g-l; 5e-f, h-m; 6b, d-f, h-i; EV 1b; EV 2a-b; EV 3c-d; EV 4b, d; EV 5b, d,
- indicate the statistical test used for data analysis in the legends of figures 1a-b, f,
- define box plots in terms of minima, maxima, centre, bounds of box and whiskers, and percentile in the legends of figures 2b-d, f-h; 4e, g-l; 5h-m; EV 2a-b,
- define n in the legend of figure 1d,
- describe the nature of entity for 'n' in the legends of figures 1h; EV 1b; EV 3c-d,
- define error bars in the legends of figures 1d; EV 1b; EV 3c-d,
- remove the legend for Table EV1 from the manuscript file and include it above the table in Appendix PDF, and
- provide a valid email address for author Wenhua Zhang - whzhangj@njau.edu.cn.

We include a synopsis of the paper (see <http://emboj.embojpress.org/>). Please provide me with a general summary image, a two-sentence summary statement and 3-5 bullet points that capture the key findings of the paper.

I look forward to receiving these changes. EMBO Press is an editorially independent publishing platform for the development of EMBO scientific publications.

Best wishes,

William

William Teale, PhD
Editor
The EMBO Journal
w.teale@embojournal.org

See also figure legend guidelines: <https://www.embojpress.org/page/journal/14602075/authorguide#figureformat>

- a point-by-point response to the referees' comments, with a detailed description of the changes made (as a word file).

- a word file of the manuscript text.
 - individual production quality figure files (one file per figure)
 - a complete author checklist, which you can download from our author guidelines (<https://www.embopress.org/page/journal/14602075/authorguide>).
 - Expanded View files (replacing Supplementary Information)
- Please see out instructions to authors
<https://www.embopress.org/page/journal/14602075/authorguide#expandedview>
- a Reagents and Tools Table as part of the Methods section, which can be downloaded from our author guidelines (<https://www.embopress.org/page/journal/14602075/authorguide#structuredmethods>)

We realize that it is difficult to revise to a specific deadline. In the interest of protecting the conceptual advance provided by the work, we recommend a revision within 3 months (27th Feb 2025). Please discuss the revision progress ahead of this time with the editor if you require more time to complete the revisions. Use the link below to submit your revision:

Referee #1:

The findings reported add new layers of biological complexity (natural variation) and of protein regulation (substrate specificity modulated by protein phosphorylation) to NRT/NPF proteins, which are key players in plant nutrition. The findings are also of potential interest to plant biotechnologist as a tool to reduce Cl/NO₃ antagonism.

Authors have addressed all my queries satisfactorily either by conducting additional research or improving clarity in the main text and figures.

At this point I can only congratulate them for this neat piece of work.

Referee #2:

I have no more critical comments about this MS

All editorial and formatting issues were resolved by the authors.

Dear Prof. Zhang,

I am pleased to inform you that your manuscript has been accepted for publication in the EMBO Journal.

Congratulations on a really convincing set of experiments!

Best wishes,

William

William Teale, PhD
Editor
The EMBO Journal
w.teale@embojournal.org
